# The Killer’s Web: Interconnection between Inflammation, Epigenetics and Nutrition in Cancer

**DOI:** 10.3390/ijms25052750

**Published:** 2024-02-27

**Authors:** Marisabel Mecca, Simona Picerno, Salvatore Cortellino

**Affiliations:** 1Laboratory of Preclinical and Translational Research, Centro di Riferimento Oncologico della Basilicata (IRCCS-CROB), 85028 Rionero in Vulture, PZ, Italy; marisabel.mecca@crob.it (M.M.); simona.picerno@crob.it (S.P.); 2Laboratory of Preclinical and Translational Research, Responsible Research Hospital, 86100 Campobasso, CB, Italy; 3Scuola Superiore Meridionale (SSM), Clinical and Translational Oncology, 80138 Naples, NA, Italy; 4S.H.R.O. Italia Foundation ETS, 10060 Candiolo, TO, Italy

**Keywords:** inflammation, epigenetic, DNA repair, nutrition, cancer

## Abstract

Inflammation is a key contributor to both the initiation and progression of tumors, and it can be triggered by genetic instability within tumors, as well as by lifestyle and dietary factors. The inflammatory response plays a critical role in the genetic and epigenetic reprogramming of tumor cells, as well as in the cells that comprise the tumor microenvironment. Cells in the microenvironment acquire a phenotype that promotes immune evasion, progression, and metastasis. We will review the mechanisms and pathways involved in the interaction between tumors, inflammation, and nutrition, the limitations of current therapies, and discuss potential future therapeutic approaches.

## 1. Introduction

Cancer is a complex disease that is influenced by both intrinsic and extrinsic factors. The accumulation of genetic mutations is a key trigger in the development and progression of tumors, but it is not sufficient on its own. In the early stages of cancer, the immune system acts as a guardian and limits tumor growth. However, tumor cells can develop mutations that allow them to evade recognition by the immune system, leading to their expansion and growth (immunoediting theory) [1]. To evade the immune response, tumor cells use various strategies. Recently, it has been discovered that the genetic instability of tumor cells induces an inflammatory response due to the release of nuclear DNA into the cytoplasm, caused by DNA repair defects [2,3]. One of the contributing factors to cancer progression is persistent inflammation, which is accompanied by the production of proinflammatory cytokines that recruit myeloid cells to the tumor microenvironment. The myeloid cells then differentiate into immunosuppressive cells, thereby fostering an environment that encourages tumor growth [4]. In addition, inflammation contributes to the epigenetic remodeling of both the tumor cells and the immune system, leading to the exhaustion of T lymphocytes and the acquisition of a malignant and aggressive phenotype by the tumor cells [5]. In this context, nutrition can play a crucial role in preventing tumor onset, reducing inflammation, and influencing the genetic and epigenetic reprogramming of tumor cells [6]. The present review endeavors to elucidate the influence of inflammation, nutrition, and epigenetic modifications on tumor progression, with a particular emphasis on the pathways that are regulated by these mechanisms and how they can be targeted by pharmaceutical agents.

## 2. Inflammation

Inflammation is a highly intricate process that is regulated by the immune system in response to external or internal stimuli that threaten tissue integrity [7]. The innate immune system is primarily responsible for triggering this response. It recognizes molecules or portions of molecules that are released upon cellular stress or tissue injury, such as damage-associated molecular patterns (DAMPs), or are specific to pathogens, such as pathogen-associated molecular patterns (PAMPs). PAMPs and DAMPs bind to pattern recognition receptors (PRRs) located on the cytoplasmic or endosome membrane, activating intracellular signaling cascades that result in the expression of proinflammatory factors [8]. PRRs include toll-like receptors (TLRs), retinoic acid-inducible gene 1-(RIG1)-like receptors (RLRs), cytosolic DNA sensor cyclic GMP–AMP synthase (cGAS) stimulator of interferon genes (STINGs), C-type lectin receptors (CLRs), and nucleotide-binding oligomerization domain (NOD)-like receptors (NLRs) (Figure 1).

TLRs, transmembrane proteins, are located on the cytoplasmic membrane or intracellular compartments. They recognize PAMPs or DAMPs, recruit MyD88 and TRIF adapter proteins, and activate the signaling cascade, leading to nuclear factor kappaB (NF-κB), interferon regulatory factors (IRFs), or mitogen-activated protein kinase (MAPK) activation and cytokines, chemokines, and type I interferon (IFN) expression [9] (Figure 1).

The RLR family encompasses melanoma differentiation-associated factor 5 (MDA5), laboratory of genetics and physiology 2 (LGP2), and RIG-1 [10,11]. Upon engagement with double-stranded RNA (dsRNA) or 5′-triphosphate single-stranded RNA, MDA5, LGP2, and RIG-I bind to mitochondrial antiviral signaling proteins (MAVS), activating interferon regulatory factor 3 (IRF3) and 7 (IRF7) through TANK-binding kinase 1 (TBK1) and IκB kinase ε (IKKε) [10,12]. IRF3 and IRF7, in combination with NF-κB and activator protein-1 (AP-1), promote the transcription of type I IFNs and other antiviral or immunoregulatory genes [10,12]. Unlike MDA5, RIG-I also binds to double-stranded viral DNA and leads to the expression of type I IFNs through the transcription of IRF3 [13]. The RIG-I-mediated inflammatory response can also be induced by lipopolysaccharide (LPS), interferon gamma (IFN-γ), interleukin (IL)-1β, and tumor necrosis factor-alpha (TNF-α) [14,15,16,17] (Figure 1).

The cGAS-STING signaling pathway plays a crucial role in mediating the inflammatory response to infections, cellular stress, and tissue damage by binding to pathogenic and nonpathogenic DNA [18]. Upon binding to dsDNA, activated cGAS synthesizes 2′3′ cyclic GMP-AMP (cGAMP), which promotes the translocation of STING from the ER membrane to the ER-Golgi intermediate and Golgi compartments [19]. STING then forms a complex with TBK1, which activates IRF3 and NFkB, leading to the transcription of dozens of genes encoding inflammatory cytokines such as IL6 and IL12, as well as antiviral type I interferons [20,21,22] (Figure 1).

CLRs are soluble and transmembrane proteins expressed by myeloid cells, particularly by macrophages and dendritic cell (DC) subsets, which are divided into two groups: the mannose receptor family [23,24] and the asialoglycoprotein receptor family [25,26]. CLRs recognize mannose, fucose, and glucan structures expressed by pathogens [27,28] and promote pathogen internalization and degradation, leading to subsequent antigen processing and presentation [28]. CLRs modulate cytokine production by inducing multiple signal transduction cascades mediated by immunoreceptor tyrosine-based activation motif (ITAM)-containing adapter molecules such as the Fc receptor γ chain (FcRγ), or by activating protein kinases such as RAS and RAF and phosphatases such as SHP1 or SHP2 [25,29]. Some CLRs regulate Toll-like receptor (TLR)-induced gene expression at either the transcriptional or post-transcriptional level, while others activate cytokine expression independently of PRR signaling pathways. In general, CLRs trigger signal transduction by controlling NF-kB function (Figure 1).

NLRs form a family of PRR proteins, characterized by the presence of a C-terminal leucine-rich repeat (LRR), a central nucleotide-binding domain (NOD), and an N-terminal effector domain [30]. Based on their N-terminal structure, NLRs are classified into five subgroups: NLRA, NLRB, NLRC, NLRP, and NLRX [31]. Since only some NLRs, such as NOD1 and NOD2, recognize and bind specific microbial products, NLRs serve as modulators of the TLR, RLR, and CLR signaling pathways. NOD1 and NOD2 activate NF-kB by interacting with RIP2, a receptor-interacting protein kinase. The activation of NF-kB by NOD1 occurs through the formation of a transient NOD1-RIP2-IKK complex, while NOD2 activates NF-kB via RIP2-dependent ubiquitination of NEMO, resulting in the release of pro-inflammatory cytokines, such as TNF-a, IL-1b, and IL-6, by monocytes and dendritic cells (DCs) [32,33].

The inflammatory response involves several NLRs, including NLRP1, NLRP3, and NLRC4, which form part of the inflammasome, a multiprotein complex. The inflammasome consists of sensor proteins, such as NLRs, absent in melanoma 2 (AIM2), and pyrin, as well as scaffold proteins, including apoptosis-associated speck-like protein containing a CARD (ASC) and the cysteine protease procaspase-1 [34,35]. The expression of pro-interleukin-1β (pro-IL-1β), pro-IL-18, and nod-like receptor protein 3 (NLRP3) is regulated by NF-κB, which primes inflammasome activation. In the canonical pathway, sensor proteins bind to DAMPs and PAMPs, leading to oligomerization and the recruitment of ASC to form a multimeric complex called a “speck”. The ASC speck complex promotes the recruitment and activation of caspase 1, resulting in inflammasome activation and the conversion of pro-IL-1β and pro-IL-18 into active IL-1β and IL-18. The inflammasome also cleaves the precursor of Gasdermin D (GSDMD), releasing the amino-terminal domain of GSDMD, which forms a pore on the plasma membrane. Mature IL-1β and IL-18 are released through this pore [36,37,38,39,40]. The secondary self-cleavage of CAS1 inactivates its enzymatic activity and inhibits inflammasome function [41,42].

In addition to the canonical caspase 1-dependent pathway, the inflammasome can be activated through caspase 4- and 5-dependent pathways, known as the non-canonical pathway. This non-canonical caspase-4/5 inflammasome is activated by intracellular LPS and requires NLRP3 for its activation. It plays a crucial role in supervising cytosolic Gram-negative bacteria by sensing lipopolysaccharide (LPS). Extracellular LPS generated by Gram-negative bacteria is internalized via TLR4- or RAGE/HMGB1-mediated endocytosis. Subsequently, guanylate-binding proteins (GBPs), which are expressed in response to interferons and other proinflammatory cytokines, alter the integrity of the endosomal membrane, thereby promoting the release of LPS into the cytoplasm. Cytosolic LPS binds directly to caspase 4–5, which then dimerizes to the active form and leads to the cleavage of GSDMDs and activation of the canonical NLRP3 inflammasome [43].

Inflammasome activation can have mild or severe consequences on cell fate. Mild activation of the inflammasome allows the endosomal sorting complex required for transport (ESCRT) machinery to remove GSDMD pores and repair the plasma membrane, whereas robust activation of the inflammasome can generate a number of pores, exceeding the capacity of the ESCRT machinery to resolve them. This imbalance leads to irreversible damage and lytic cell death, such as pyroptosis, a form of programmed inflammatory cell death dependent on the activation of inflammasomes and initiated by microbial infections or various pathological stimuli, such as stroke, heart attack, or cancer [41,42].

Inflammasome activation triggers the release of proinflammatory cytokines and DAMPs, including high-mobility group box 1 (HMGB1). HMGB1 can be released into the extracellular space both actively secreted by inflammatory and immune cells and passively secreted by damaged or necrotic cells with compromised plasma membranes. These proinflammatory cytokines and DAMPs act as chemoattractants for neutrophils, macrophages, monocytes, dendritic cells, and T cells. They also regulate the adaptive response by finely orchestrating a series of mechanisms involved in the production of proinflammatory and anti-inflammatory cytokines, thereby preventing tissue damage caused by hyperactivation of the immune system.

The activation of inflammasomes results in the production of pro-inflammatory cytokines, such as IL-1α, IL-1β, and IL-18, as well as the maturation of anti-inflammatory cytokines like IL-37. IL-1α is responsible for inducing the release of pro-inflammatory cytokines such as TNFα and IL-2, and acts as a chemoattractant signal for neutrophils. IL-1β stimulates inflammatory responses by releasing pro-inflammatory cytokines such as IL-6 and IL-17a, as well as recruiting macrophages [44]. The pro-inflammatory cytokine IL-18 enhances adaptive immune responses by regulating leukocyte trafficking via chemokine production and promoting the secretion of proinflammatory cytokines IFNγ, IL-2, and IL-12 [45,46]. IL-37, secreted in response to inflammatory signaling, downregulates the production of pro-inflammatory cytokines to restrain immune cell toxicity and prevent tissue damage [47].

Generally, organisms possess intricate systems that identify and eradicate pathogens, protect against external infections, and maintain tissue homeostasis. The activation of these systems triggers an acute inflammatory response that involves the remodeling of local tissue through the release of cytokines that attract immune cells and regulate their functions. Acute inflammation is beneficial because it can quickly resolve tissue damage caused by pathogens or internal stimuli. However, if acute inflammation fails to eliminate the causes of inflammation and repair tissue damage, the inflammatory response persists, becomes chronic, and develops new characteristics that can damage DNA and compromise tissue health [48]. Chronic inflammation does not appear to be caused solely by infections or injuries but, rather, by malfunctions in tissue due to factors, such as DNA repair deficiency, nutrition, epigenetic remodeling, and cancer.

## 3. DNA Damage Response in Inflammation

### 3.1. Mechanism Underlying DNA Damage-Induced Inflammation

DNA repair mechanisms and signaling pathways are closely linked to the inflammatory response. DNA damage activates the cGAS-STING pathway, which induces the expression of type I interferons and inflammatory factors. cGAS triggers this process by detecting endogenous DNA released from the nucleus, mitochondria, or micronuclei into the cytoplasm as well as potentially exogenous DNA derived from pathogenic microorganisms [49,50,51]. Micronuclei, which are small, membrane-enclosed structures composed of acentric chromosomal fragments, are generated as a result of defects in DNA repair processes, chromosomal segregation, and non-disjunction. These defects can be caused by mutations in genes involved in the DNA damage response (DDR) and the regulation of the mitotic spindle, or they can be induced by DNA-damaging treatments [52,53,54]. Unrepaired or misrepaired DNA double-strand breaks (DSBs) lead to the formation of acentric chromatids that segregate improperly during anaphase, are excluded from the nucleus, and are enveloped by the nuclear membrane [52]. In 60% of micronuclei, the nuclear membrane breaks down during interphase, causing disruption that leads to the recruitment and unrestrained accumulation of ESCRT-III, responsible for maintaining and preserving the integrity of the nuclear envelope. The accumulation of ESCRT-III deforms the nuclear membrane, promoting DNA damage, chromosome instability, and DNA leakage into the cytoplasm [55]. The migration of chromosomal DNA fragments from micronuclei to the cytoplasm results in the activation of cGAS, which subsequently leads to the synthesis of cGAMP. This, in turn, triggers the inflammatory process by inducing the production of type I interferon and cytokines via the STING-TBK1-IRF3 and STING-TBK1-NFkB signaling pathways, respectively [20,56].

Although the c-GAS-STING pathway serves as the primary mechanism to induce inflammatory responses to genotoxic stress, DDR proteins can also trigger inflammation by directly or indirectly activating NF-kB. Damaged DNA, induced by genotoxic agents, recruits and activates ATM and ATM and RAD3 related (ATR), which can stimulate NF-kB activation through the following: (1) stabilization of GATA4 through inhibition of p62 and autophagic degradation of GATA4 [57]; (2) assembly of an alternative STING signaling complex including the tumor suppressor p53 and the E3 ubiquitin ligase TRAF6 [58]; (3) degradation of IκBα through the formation of the IκBα-β-TrCP-ubiquitin ligase complex and phosphorylation of RELA (also known as p65) through interaction with protein kinase A (PKA) [59].

The protein kinase, DNA-PK, which plays a crucial role in non-homologous end joining for repairing double-strand breaks, has been found to have contrasting functions in regulating the inflammatory response. DNA-PK has been shown to suppress the inflammatory process by inhibiting the synthesis of cGAMP through the phosphorylation of cGAS [60], while also triggering innate immunity through IRF3-dependent mechanisms by acting as a cytoplasmic DNA sensor [61].

### 3.2. DNA Repair Deficiency Disorder and Inflammation

The DDR is a vital mechanism that ensures the preservation of genomic integrity and prevents tumor formation by employing an intricate signaling network that detects, signals, and repairs DNA damage. Numerous recent investigations have pointed out that the deletion or mutation of DDR genes, both inherited and spontaneous, as well as the genotoxic stress induced by DNA-damaging treatments, can trigger inflammation by activating the cGAS-STING pathway and other associated signaling cascades [49,62,63,64,65,66].

For example, hereditary mutations in the ataxia-telangiectasia (ATM) gene, a DNA damage sensor protein, result in a genetic disorder that affects the nervous and immune systems, leading to intense pathological inflammation and elevated levels of IL-6 and IL-8. Individuals diagnosed with ataxia-telangiectasia experience an accumulation of cytosolic DNA, which results from unrepaired double-strand breaks in genomic DNA. This accumulation activates the cGAS-STING and AIM2 inflammasome pathways, leading to the upregulation of the type I and type III interferon signaling pathways [49,67,68].

The deficiency of BRCA1 or BRCA2 tumor suppressors, which play a crucial role in repairing DNA breaks through the promotion of the homologous recombination (HR) DNA repair pathway, and in maintaining the stability of newly synthesized DNA strands by safeguarding stalled replication forks from degradation, also triggers type I IFN signaling and anti-tumor immunity. Cells lacking BRCA1 or BRCA2 exhibit elevated levels of cytosolic DNA and the constitutively active viral response cGAS/STING/TBK1/IRF3 pathway [65,66].

The c-GAS-STING pathway can also be activated by other proteins involved in DNA repair. For instance, the inactivation of BLM, a DNA helicase involved in the repair of DNA double-strand breaks through the homologous recombination (HR) pathway, can compromise the integrity of DNA by blocking the restart of stalled replication forks [69], as well as DNA double-strand break resection [70] and Holliday junction dissolution. BLM mutations are responsible for Bloom syndrome, a genetic disorder characterized by genetic instability and an increased risk of cancer. Recently, it was discovered that BLM mutations in the blood of Bloom syndrome patients trigger the activation of the c-GAS-STING pathway, type II IFN responses, and the induction of ISGs. This occurs due to the accumulation of micronuclei and cytosolic DNA fragments [64].

Studies indicate that the inactivation of the DNA excision repair protein known as ERCC-6, which is mutated in Cockayne syndrome, a disorder characterized by neurodegeneration, results in a significant inflammatory response to UV radiation in mice. This response is triggered by the disruption of the stability of mitochondrial transcription factor A (TFAM) caused by the loss of ERCC-6. As a result, mitochondrial DNA becomes fragmented and is released into the cytoplasm, where it binds and activates cGAS [71].

Even the inhibition of proteins involved in base excision repair (BER) can elicit an interferon response through activation of the cGAS pathway. Mutations in the DNA polymerase beta (POLB) protein, which is involved in BER and fills in single-nucleotide gaps, result in replication-associated genomic instability and inflammation-associated carcinogenesis in mice. This occurs because failure to repair the damage leads to the formation of single-strand breaks (SSBs), which are converted into double-strand breaks (DSBs) during the S-phase of DNA replication. The accumulation of DSBs causes mitotic dysfunction, leading to an increase in micronuclei formation due to the mis-segregation of broken chromosomes during mitosis. This, in turn, triggers a c-GAS-mediated inflammatory response by releasing cytosolic DNA [72].

Another example is the protein SAMHD1, which functions as a deoxyribonucleotide triphosphate triphosphohydrolase and plays a crucial role in the progression of DNA replication. SAMHD1 has also been implicated in several diseases, including cancer and the rare congenital inflammatory disorder Aicardi–Goutières syndrome, which is characterized by the excessive production of type I interferons (IFNs). SAMHD1 promotes the degradation of newly synthesized DNA at stalled replication forks by stimulating the activity of the MRE11 exonuclease, a protein involved in repairing double-strand breaks. The inhibition of SAMHD1 expression leads to replication failure, resulting in the accumulation of single-stranded DNA fragments in the cytosol, which are released from stalled forks, and the activation of the cGAS-STING pathway [63].

### 3.3. Inflammation Promoted by Transposable Element (TE) Instability

Dysfunctions in the DNA damage response pathway can result in a strong inflammatory response by accelerating the activation of transposable elements (TEs), which are DNA sequences capable of self-replication and relocating within the same genome [73]. The eukaryotic genome is primarily composed of repetitive DNA sequences, including satellite DNA and TEs. TEs are divided into two groups: retrotransposable elements (RTEs), such as long interspersed nuclear elements (LINEs) and short interspersed nuclear elements (SINEs), and endogenous retroviruses (ERVs) [74,75,76]. Active RTEs produce cytosolic DNA (cDNAs) that activate the cGAS-STING pathway and initiate the production of pro-inflammatory cytokines, such as IFNα, IFNβ, IL-6, and TNF, through NF-κB and IRF3 stimulation. Although the mechanism underlying RTE-induced inflammation remains unclear, it is believed that abnormal DNA damage repair mechanisms can result in the accumulation and dispersion of cDNA in the cytoplasm by affecting the integration of RTE DNA into the genome. Additionally, genomic instability caused by RTEs may cause the formation of micronuclei, which could activate the ATM and DDR signaling pathway, ultimately triggering an immune response [77].

### 3.4. Inflammation Triggered by R-Loops Resolving Defects

Inflammation can arise from disruptions in the mechanisms responsible for maintaining and resolving R-loops. R-loops are nucleic acid structures comprising DNA-RNA hybrids and displaced single-stranded DNA resulting from transcription and replication [78]. If these structures persist for an extended period, endonucleases cut the exposed single-stranded DNA, leading to single- and double-strand breaks. Alterations in helicases, such as SENATAXIN, BLM, and WRN, and endonucleases, such as ERCC1 and XPG, can impair R-loop resolution [79,80,81,82,83]. The loss of tumor suppressors BRCA1 and BRCA2, which are involved in repairing double-strand breaks, can contribute to an abnormal accumulation of R-loops and cytosolic DNA, resulting in a heightened inflammatory response [84,85].

### 3.5. Anti-Tumor Cytotoxicity Mediated by Inflammatory Response

The effectiveness of chemotherapeutic agents, including anthracyclines, oxaliplatin, and doxorubicin, depends, in part, on their ability to induce anti-tumor immune responses through type I IFN signaling and ISGs [86,87]. Similarly, the anti-tumor immunity triggered by ionizing radiation (IR) is dependent on the cGAS-STING pathway [88]. Recent studies have identified the accumulation of cytosolic DNA following chemotherapy or radiation as the trigger for cGAS-STING activation, leading to the induction of IFN and ISGs [53,54,89]. Cytosolic DNA is generated as a result of double-strand break repair processes triggered by ionizing radiation or chemotherapy. Specifically, enzymes such as BLM helicase and exonuclease 1 (EXO1), which play a crucial role in the final resection of DNA during double-strand break repair, generate these DNA fragments that can either be enclosed in micronuclei or migrate to the cytoplasm, where they activate the cGAS-STING pathway [54,89]. Conversely, the cytoplasmic 3′–5′ exonuclease Trex1 prevents the activation of the inflammatory response by degrading these fragments [90].

Inhibiting or deactivating DDR pathways can enhance the inflammatory response and improve the immune response induced by genotoxic stress agents. PARP inhibitors (PARPi) increase endogenous genomic instability in BRCA1/2-deficient cells and tumors by blocking DNA repair mechanisms and replication fork progression, which leads to an increase in cytosolic DNA and activates the cGAS-STING pathway [91,92,93]. As a result, ISGs are expressed in BRCA1/2-deficient tumor cells and stimulate T-cell infiltration and activation, ultimately leading to tumor eradication in BRCA1/2-deficient mouse models of ovarian and breast cancer [94].

Similarly, PARPi can cause DNA damage and promote genomic instability in cells and tumors deficient in ERCC1, a protein involved in the process of excision repair, and POLB-deficient cells. This instability can result in the formation of micronuclei, which, in turn, lead to the accumulation of cytoplasmic DNA. This accumulation activates the cGAS-STING pathway and triggers a type I interferon response. The activation of type I IFN signaling then elicits specific tumor cell-intrinsic immune responses [62,72]. Notably, the release of damaged DNA fragments and the accumulation of micronuclei in the cytosol, which are induced by radiotherapy, stimulate the production of IFN-I via the cGAS pathway. This, in turn, promotes the maturation of dendritic cells (DCs), enhancing their cross-priming capacity and ultimately leading to the activation of T cells [95].

Ultimately, the DDR pathway serves as a guardian of genomic integrity, encompassing not only DNA repair processes but also the activation of the inflammatory response. This latter mechanism may be employed to eliminate severely damaged or dysfunctional cells. Nevertheless, the activation of inflammation in pathological contexts, such as cancer, can have detrimental consequences, as it may adversely affect the cells in the microenvironment by altering the cellular metabolism and epigenetics.

## 4. Epigenetics and Inflammation

Epigenetics play a crucial role in regulating the inflammatory response and reprogramming tumor cells by promoting both immune evasion and drug resistance. Epigenetic mechanisms include reversible post-translational modifications (PTMs) and non-coding RNAs that remodel the chromatin structure and regulate the accessibility of transcription factors or repressors to gene promoters. Epigenetic remodeling of chromatin occurs through epigenetic modulators that modify the histone methylation or acetylation state, DNA methylation patterns, and through the expression of microRNAs (miRNAs) and long non-coding RNAs (lncRNA) that influence gene transcription and protein synthesis.

### 4.1. Epigenetic Regulatory Mechanisms

Epigenetic modulators are divided into (1) writers, which add covalent modifications to histones or DNA; (2) erasers, which remove histones and DNA modifications; and (3) readers, which recognize and bind epigenetic modifications [96,97,98] (Figure 2).

Histone modifications. Histone acetylation occurs on lysine and arginine residues and is carried out by histone acetyl transferases (HATs), which act as epigenetic regulators. The addition of an acetyl group neutralizes the positive charge of lysine and weakens the bond between the histone and negatively charged DNA, thus allowing for more open chromatin and more accessible transcription factors [99]. Histone deacetylases (HDACs) counteract the function of HATs by removing acetyl groups from histones, thus promoting chromatin compaction and silencing gene expression. HDACs include four classes; classes I, II, and IV are zinc-dependent, whereas class III HDACs, also known as sirtuins, depend on nicotinamide adenine dinucleotide (NAD^+)^ [100] (Figure 2).

Histone lysine methyl transferases (KMTs) label histones by adding mono-/di-/or tri-methyl groups to the lysine residue, thus generating a histone methylation pattern known as histone codes [101]. Histone methylation can repress or activate gene transcription, depending on the position of the methylated lysine. The trimethylation of lysine 27 on histone 3 (H3K27me3) is known to result in the silencing of gene expression, whereas the dimethylation of lysine 4 on histone 3 (H3K4me) promotes gene transcription by increasing DNA accessibility [102]. Histone-specific demethylases, like lysine-specific histone demethylase (LSD)1 and Jumonji-C (JMJC) families, remove methyl groups from histones and modify the chromatin structure, promoting epigenetic plasticity and gene transcription modulation in response to internal and external stimuli [103] (Figure 2).

In addition to methylation, histones can undergo other modifications, such as lactylation, crotonylation, succinylation, propionylation, malonylation, butyrylation, glutarylation, and 2-hydroxyisobutyrylation, which can influence their function. For instance, crotonylation and succinylation of histones have been linked to the activation of gene transcription [104].

Histone acetylation and methylation play a significant role in chromatin marking, controlling the accessibility of chromatin to epigenetic readers, such as bromodomain and extra-terminal domain (BET) proteins. BET proteins bind to acetylated histones via the bromodomain and enhance transcriptional activation by facilitating chromatin opening and recruiting coactivators and transcription factors to gene promoters [105]. Similarly, histone methylation readers bind histones through specific methyl-binding motifs, such as PHD, chromium, tudor, PWWP, WD40, BAH, ADD, ankyrin repeat, MBT, and zn-CW domains. These proteins are specifically recruited by histones based on their methylation status and the amino acid sequence surrounding the methylation sites, and they can recruit either repressors or activators of transcription depending on the histone methylation signature [106]. However, to initiate transcription, an additional epigenetic modification is necessary, specifically DNA demethylation (Figure 2).

DNA methylation. DNA methylation is a mechanism of epigenetic regulation that involves the transfer of a methyl group from S-adenosyl methionine (SAM) to cytosine on carbon 5, which is catalyzed by DNA methyltransferases (DNMTs), such as DNMT1, DNMT3A, and DNMT3B. This results in the formation of 5-methyl-cytosine (5-mC) [107]. Cytosine methylation within CpG islands, which are genomic regions composed of multiple repeats of CpG dinucleotides positioned upstream of the promoter, can repress gene transcription by preventing the recruitment of transcription factors or promoting the binding of repressor complexes to methylated DNA [108]. However, CpG islands in gene promoters only make up a small part of the genome. The majority of CpG dinucleotides are found in repetitive DNA, known as repetitive elements (REs), which are the most abundant sequence type in the genome [109]. The methylation status of these REs is essential for genome stability, replication, regulation of gene transcription, and nuclear architecture [110,111,112,113]. Demethylation is mediated by the activity of ten-eleven translocation (TET) proteins and thymine-DNA glycosylase (TDG). TETs convert 5-methyl-cytosine (5-mC) into 5-hydroxymethylcytosine (5-hmC), 5-formylcytosine (5-fC), and 5-carboxylcytosine (5-caC) [114,115]. TDG removes 5-fC and 5-caC paired with guanosine, and the subsequent base excision repair (BER) machinery restores the G:C pair [116]. The maintenance of the genomic methylation state also depends on proteins such as methyl CpG-binding protein 4 (MBD4), which contains both a methylated DNA-binding domain (MBD) and a DNA glycosylase domain, to remove thymine mismatched to guanosine [117]. MBD4 binds 5-mC via its MBD and removes thymine derived from the deamination of 5-mC, while the BER machinery completes the repair by inserting an unmodified cytosine. MBD4 then recruits DNMT1 and promotes the remethylation of newly repaired CpG sites, ensuring the maintenance of the genomic methylation pattern [118]. Post-translational modifications of proteins involved in epigenetic regulation, such as ubiquitination and SUMOylation, can affect their function and impair epigenetic mechanisms [119,120] (Figure 2).

Transcriptional regulation is tightly controlled by the coordinated action of histone and DNA modification proteins. Methylated DNA interacts with the methyl-CpG-binding protein (MBD) family, which facilitates histone deacetylation by attracting various transcriptional corepressor complexes that contain HDACs [121]. DNMT1 and DNMT3a limit gene expression by binding to SUV39H1, an enzyme that methylates H3K9 [122]. Additionally, DNMT1 and DNMT3b interact with HDACs and regulate gene expression [123,124]. While the precise mechanism governing the interaction between histone acetylation and DNA demethylation remains elusive, it is worth noting that two studies have revealed that suppressing HDAC triggers active DNA demethylation [125,126].

Non-coding RNA. MiRNAs are double-stranded RNA oligonucleotides that regulate gene expression by interfering with mRNA transcription and translation. Long non-coding RNAs (lncRNAs) are a class of transcripts with little or no protein-coding capacity that can originate from intergenic regions, intronic regions, or exonic regions and can be either sense or antisense in their orientation relative to protein-coding genes. LncRNA transcripts are longer than 200nt and regulate chromatin architecture and transcription, splicing, protein translation, and localization through RNA-RNA, RNA-DNA, and RNA-protein interactions [127,128,129] (Figure 2).

### 4.2. Epigenetic Signatures Underlying Inflammation

In this section, we will examine the alteration of various epigenetic mechanisms associated with inflammatory diseases.

Histone acetylation. Histone acetyltransferases (HATs) and histone deacetylases (HDACs) regulate the expression of several genes involved in the inflammatory response [130]. NF-κB-dependent gene expression requires the involvement of histone acetylase, which will decompress the repressed chromatin environment through histone acetylation [131,132]. Several inflammatory lung diseases are associated with increased H3 acetylation in a specific promoter region that is regulated by NF-κB. Histone modifications at specific acetylation sites (H3K9ac, H3K14ac, H3K18ac, H3K23ac, H3K27ac, H3K36ac, H2B1KK120ac, H2B2BK20ac, H2BK16ac, H2BK20ac, H2BK108ac, H2BK116ac, and H2BK120ac upregulation; H2BK5, H2BK11 downregulation) are involved in the pathogenesis of asthma [133]. CBP/p300 acetyltransferase promotes the transcription of various proinflammatory cytokines, such as IL-1, IL-2, IL-8, and IL-12, by acetylating histones associated with the promoter region of these genes [130]. The enhanced expression of TNF-α upon LPS stimulation is associated with epigenetic remodeling of the TNF-α locus, which is characterized by increased histone 4 acetylation [134]. In paraquat-induced pulmonary fibrosis, elevated IL-6 expression depends on acetylation of H3K9ac in the IL6 promoter region [135]. In addition, the hyperacetylation of H3K9/K14 and H4K12 in the IL-6 promoter region causes the elevated expression of IL-6 in synovial fibroblasts from osteoarthritis patients [136].

In contrast, certain proteins like promyelocytic leukemia zinc finger (PLZF) play a role in suppressing the inflammatory response triggered by the activation of TLR- and TNFα-dependent signaling pathways. PLZF does so by promoting chromatin remodeling, which, in turn, represses the NF-kB response by ensuring the stability of a co-repressor complex that comprises HDAC3 histone deacetylase and the NF-kB p50 subunit [137].

Histone Methylation. The regulation of epigenetic histone methylation is crucial in the inflammatory response and has a significant impact on the transcription of various genes, including those involved in the production of pro-inflammatory cytokines. The histone demethylase LSD1 has various effects on the expression of pro-inflammatory cytokines depending on the cell type and immune signals. In smooth muscle cells (SMCs) and hematopoietic stem cells (HSCs), LSD1-mediated demethylation of H3K4me2 suppresses cytokine gene expression but triggers an inflammatory response in conditions such as rheumatoid arthritis (RA) and sepsis [138,139]. In contrast, the demethylation of H3K27me2 by JMJD3 allows for the recruitment of p65 to the Il11 promoter, leading to the expression of IL-11 and other cytokines including IL-1β and TNF-α [140]. Exposure to inflammatory cytokines activates JMJD3, promoting the transcription of polycomb-group protein (PcG) target genes by removing H3K27me3-repressive marks from the promoter region [141]. In Il4-treated macrophages, JMJD3 removes H3K27me3-repressive marks from the STAT6 promoter and stimulates the transcription of specific inflammatory genes [142]. KDM7C regulates proinflammatory cytokines, like TNF, CCL4, and Il11, upon TLR-4 activation by LPS. TLR-4 stimulates NF-kB, which then recruits and delivers JHDM1E to the promoter region of these genes, where it demethylates H4K30me3 to initiate transcription [143]. KDM6A and KDM7A are critical for promoting endothelial inflammation caused by TFN-α. They accomplish this by demethylating H3K27me3 in the promoter regions of ADAM17 and Jagged-1, which are responsible for initiating the inflammatory process [144].

Histone methyltransferases also suppress PRR signal transduction. Following papilloma virus infection, the E7 oncoprotein forms a complex with NF-κBp50–p65 and recruits histone demethylase JARID1B and histone deacetylase HDAC1. This complex binds to a specific region on the TLR9 promoter, leading to decreased methylation and acetylation of histones upstream of the TLR9 transcriptional start site [145]. The overexpression of KDM5B/5C histone demethylases has been found to have a detrimental impact on the innate immune response in breast cancer. Specifically, it has been observed to inhibit the STING transcription through the demethylation of H3K4me3 at the STING locus [146].

DNA methylation. The methylation status of CpG islands significantly affects the inflammatory response and the progression of various inflammatory diseases. In systemic lupus erythematosus (SLE), the overexpression of CD11a, CD40L, CD70, KIR2DL4, and PRF1 in T lymphocytes is associated with DNA hypomethylation in promoter regions [147]. In pulmonary inflammatory diseases such as asthma and chronic obstructive pulmonary disease (COPD), hypomethylated CpG patterns are correlated with disease severity. Specific DNA methylation signatures are present in ulcerative colitis patients [148]. DNA hypermethylation in the promoters of CXCL14, CXCL5, GATA3, IL17C, and IL4R and hypomethylation in the promoters of CCL25, IL13, IL12B, CXCL5, IL4R, IL17A, and IL17RA are associated with the onset or aggravation of ulcerative colitis [149]. In autoimmune diseases, the expression of inflammasome genes is dependent on the methylation status of their promoters. Monocytes from patients with chronic non-bacterial osteomyelitis (CNO) express high levels of inflammasome genes, such as NLRP3, because the promoters of the respective genes are demethylated [150].

Non-coding RNA. Several microRNAs (miRNAs), including miR-126, miR-132, miR-146, miR-155, and miR-221, have been implicated in the regulation of inflammatory genes [151]. miR-10a, in particular, has been shown to suppress the expression of inflammatory cytokines, such as IL-6, IL-8, MCP1, MMP1, and MMP13, in fibroblast-like synoviocytes (FLSs), thereby preventing the release of proinflammatory interleukins and chemokines [152].

Long non-coding RNAs (lncRNAs) also play a role in modulating the inflammatory response by regulating the ubiquitination, stability, and activity of proteins and remodeling chromatin. For instance, the MaIL1 lncRNA and lncRNA NRIR have been shown to promote IFN-I production by activating the ISG pathway upon LPS stimulation [153]. Other lncRNAs, on the other hand, act as suppressors of IFN, cytokines, and chemokine production. lnc-Lsm3b, for example, inhibits the IFN signaling pathway by binding and blocking RIG-I, while lincRNA-EPS inhibits cytokine and chemokine expression by promoting a repressive chromatin state at the promoters of cytokine and chemokine genes [153,154].

The activation of the TLR-NFκB signaling pathway by LPS induces the expression of the lncRNAs IL-1β-eRNA and IL-1β-RBT46, which play a positive role in the production of proinflammatory cytokines [155]. PACER and lncRNA CARLR, on the other hand, induce the expression of pro-inflammatory genes through the activation of NF-κB [156], while other lncRNAs, such as ROCKi and IL7-AS, promote the expression of TLR-MAPK/NFκB-dependent genes, such as CCL2 and IL6, favoring chromatin opening and histone acetylation at the gene promoter level [157]. In contrast, the activation of the TLR-MyD88-NFκB signaling pathway can reduce the expression of several lncRNAs that act as repressors of NFκB-dependent genes [158].

LncRNAs are also involved in the onset of several autoimmune diseases. For example, lncRNAs, such as Hotair, LUST, anti-NOS2A, MEG9, SNHG4, TUG1, and NEAT1, have been found to be highly expressed in rheumatoid arthritis, while the expression of lnc-HSFY2–3:3 and lnc-SERPINB9–1:2 is reduced in systemic lupus erythematosus (SLE) [159].

### 4.3. Epigenetic Modulation of the Immune Function

In addition to their role in the inflammatory response, epigenetic modifications are crucial in the differentiation and maintenance of T-lymphocyte functions. In the inactive or naive state, DNMT1 methylates the promoter regions of transcription factors that define the T-cell lineage, thereby suppressing their expression. Upon antigenic stimulation, TET proteins assist in removing the methyl group from gene promoters, triggering the transcription of specific transcription factors, such as T-bet, GATA-3, RORgt, and Foxp3, which are expressed in each T-cell subset [160,161]. The differentiation of T lymphocytes into specific subsets, such as Th1 and TH2, involves epigenetic modifications. For instance, helper T-lymphocyte polarization into the Th1 subset is characterized by an increase in H4 acetylation on the Ifn-γ promoter [162], while polarization into the Th2 subset results in the repression of Ifn-γ via H3K27me3 methylation by EZH2 [163]. The differentiation of CD8 T cells into effector T cells, such as cytotoxic T cells, is accompanied by increased expression of IFN-γ, perforin (PRF), and granzyme B (GZB) genes due to demethylation of their promoter regions [164,165]. LncRNAs also play a significant role in the activation and differentiation of lymphocytes. Several lncRNAs, such as NRON, NKILA, BCALM, GAS5, and PVT1, regulate the activation of T lymphocytes and the TCR/BCR signaling pathway by modulating the activity of NFAT, NFκB, and MYC [166,167,168,169,170]. Additionally, lncRNAs such as IFNG-AS1 and TH2-LCR are involved in the differentiation of T lymphocytes into various effector subsets [171,172].

Epigenetic mechanisms are crucial in regulating inflammatory responses and in the functioning and differentiation of immune cells. In pathological situations, epigenetic reprogramming can worsen physiological conditions, promoting the advancement of diseases. Nevertheless, these modifications are reversible, making them attractive therapeutic targets in a variety of diseases. The regulation of these mechanisms is impacted by both internal and external factors, one of which is diet.

## 5. Nutrition, Inflammation, and Epigenetics

### 5.1. Role of Nutrition in Inflammation

Nutrition exerts a critical influence on the inflammatory response, as it regulates diverse genetic and epigenetic mechanisms involved in this process. The inflammatory response is dependent on the activity of eicosanoid hormones, specialized pro-resolution mediators (SPMs), and modulatory genes [173].

Eicosanoids are primarily derived from arachidonic acids, which are membrane phospholipids that are released upon the activation of phospholipase A2 by an inflammatory insult. The intensity of the inflammatory response is proportional to the amount of arachidonic acid present in the membrane [174]. The production of arachidonic acid is contingent upon the activity and regulation of two enzymes, delta-6-desaturase and delta-5-desaturase, which are responsible for converting linoleic acid to arachidonic acid. Elevated levels of insulin activate these enzymes, and insulin resistance induced by cytokines, such as TNF-α, can contribute to the production of arachidonic acid [175,176]. In contrast, glucagon, secreted by the pancreas in proportion to dietary protein content, inhibits the two delta-desaturases and, consequently, the production of arachidonic acid [175]. Elevated concentrations of saturated fatty acids, particularly palmitic acid, in the blood have been found to interact with toll-like receptors TLR-2 and TLR-4, leading to inflammation through the activation of NF-kB [177]. A diet supplemented with omega-3 fatty acids, particularly the long-chain omega-3 fatty acids eicosapentaenoic acid (EPA) and docosahexaenoic acid (DHA), reduces the inflammatory state by inhibiting delta-5-desaturase and inflammasome activation [178]. Although EPA and DHA generate eicosanoids, their inflammatory activity is much lower (100–1000-times) than that of eicosanoids derived from arachidonic acid. Nonetheless, a diet rich in omega-3 fatty acids still contributes to reducing the intensity of the inflammatory response [179].

Specialized pro-resolving mediators (SPMs) are hormones derived from EPA, DHA, and docosapentaenoic acid (DPA), which play a crucial role in the resolution of residual inflammation [180]. These SPMs include resolvins, maresins, and protectins and exert their anti-inflammatory effects by inhibiting the migration of neutrophils to the site of injury, promoting the transition of macrophages from the pro-inflammatory M1 subtype to the anti-inflammatory M2 subtype and facilitating the removal of apoptotic cells by professional and non-professional phagocytes [181]. To achieve complete resolution of inflammation, it is essential that omega-3 fatty acids reach specific levels in the blood to support the production of SPMs [182].

The 5′-adenosine monophosphate-activated protein kinase (AMPK) gene is the most critical regulator of inflammation modulation, as it inhibits NF-kB. AMPK regulates cellular energy homeostasis and is activated in response to energy stress, leading to an increase in the AMP/ATP ratio. Activation of AMPK facilitates the metabolic shift from anabolism to catabolism, initiates autophagy, which is vital for supplying energy, metabolites, and biosynthetic intermediates necessary for healing inflammatory lesions, and stimulates mitophagy to replace damaged mitochondria with functional organelles capable of providing the energy required for resolving inflammatory damage [183,184,185,186]. AMPK activity is positively regulated by the SPM signaling pathway, as the binding of SPM, such as Resolvin D1 (RvD1), to SPM receptors, such as FPR2/ALX, promotes AMPK activation [187].

Polyphenols, natural compounds derived from plants, fruits, and vegetables with antioxidant properties, can reduce the inflammatory response and promote resolution of the inflammatory state through the indirect activation of AMPK. Polyphenols, particularly anthocyanins, have been demonstrated to exert a positive regulatory effect on AMPK through the activation of sirtuins. Sirtuins, in turn, stimulate the transcription of hepatic kinase B1 (LKB1), which subsequently promotes the transcription of AMPK by deacetylating the AMPK promoter [188,189]. In turn, activated AMPK supports the activity of sirtuins by promoting the regeneration of NAD^+^, activating nicotinamide phosphoribosyltransferase (NAMPT), the rate-limiting enzyme of the NAD^+^ salvage pathway [190].

However, an essential requirement for an effective anti-inflammatory diet is a low-calorie intake. Calorie restriction (CR) exerts potent anti-inflammatory effects through the activation of the anti-stress response, which leads to a reduction in oxidative stress and the protection of DNA from oxidative damage by activating AMPK and suppressing protein kinase A (PKA) and mammalian target of rapamycin (mTOR) signaling pathways, which are triggered by glucose and amino acids, such as threonine and valine [191] (Figure 3). CR and fasting have been shown to reduce the production of interleukin-1 beta (IL-1β) in monocytes in response to lipopolysaccharides (LPSs) by activating deacetylase sirtuin-3 [192]. Ketone bodies, such as acetoacetate and beta-hydroxybutyrate, generated during fasting have been shown to inhibit the NLRP3 inflammasome by binding and activating the G-protein-coupled receptor GPR109A (HCAR2), which is abundantly expressed by monocytes and macrophages [193,194].

In recent times, dietary interventions such as caloric restriction (CR) and periodic fasting (PF) have demonstrated numerous benefits, ranging from the prevention to the enhancement of treatments for human diseases [195,196,197,198,199,200,201,202,203,204]. Studies have indicated that CR and PF can prolong the lifespan of various organisms, ranging from invertebrates such as worms and flies to vertebrates including rodents and primates, by lessening the risk of chronic inflammation and postponing the emergence of age-related disease, including Alzheimer’s disease, diabetes, high blood pressure, cardiovascular disease, and cancer, which are linked to increased inflammation [198,204]. Furthermore, cycles of PF devoid of essential amino acids have been shown to alleviate the severity of multiple sclerosis, a chronic, inflammatory disease, in mouse models by promoting oligodendrocyte regeneration and reducing the levels/reactivity of pathological autoimmune cells [205].

CR and fasting have an impact on the populations of intestinal microbiota, which are influenced by the macronutrient composition and metabolome content of the diet. Clinical and preclinical studies have demonstrated that CR and fasting can enrich the microbiota with probiotic-favorable strains, such as Lactobacillus and Bifidobacterium, while simultaneously reducing pro-inflammatory strains, such as Desulfovibrionaceae and Streptococcaceae [206,207]. Consuming fermentable fiber, omega-3 fatty acids, and polyphenols can promote the growth of Akkermansia muciniphila, which, in turn, strengthens the intestinal barrier and prevents the release of bacterial endotoxins into the bloodstream, thereby reducing inflammation.

Therefore, a hypocaloric diet low in fatty acids, such as palmitic, arachidonic, and linoleic acid, and high in fiber, polyphenols, and omega-3 fatty acids can effectively combat inflammation [208,209].

### 5.2. Role of Nutrition in Epigenetic Reprogramming

Epigenetic mechanisms are influenced by nutrition, which regulates metabolic pathways and produces intermediate metabolites that serve as coactivators for the enzymes involved in the modification of chromatin and DNA. A diet low in carbohydrates and proteins, including CR and PF, causes a metabolic shift from glycolysis to ketone bodies and fatty acid-based oxidative phosphorylation (OXPHOS). This metabolic change results in alterations in the metabolites produced by glycolysis and the Krebs cycle, such as NAD^+^, flavin adenine dinucleotide (FAD^+^), lactate, and α-ketoglutarate, which impacts gene expression by regulating epigenetic writers involved in the DNA methylation and histone acetylation, ADP ribosylation, methylation, O-GlcNAcylation, and lactylation [191,196,198,210].

NAD^+^ serves as a substrate for both sirtuin deacetylases and ADP-ribosyltransferases (ARTs). By adding ADP-ribose to histone proteins, these enzymes promote chromatin relaxation and increase the accessibility of transcription complexes to genomic DNA. As a result, the increase in NAD^+^ induced by energy stress from CR leads to epigenetic remodeling and a rewiring of transcription through the deacetylation of histones by sirtuins at certain gene loci and the ADP-ribosylation of other loci, resulting in gene transcriptional reprogramming [211].

α-Ketoglutarate is a crucial metabolic intermediate of the Krebs cycle that functions as a cofactor for various chromatin-modifying enzymes, such as histone demethylases (KDMs) and the TET family of enzymes, which are involved in DNA demethylation with TDG. The metabolic changes that occur due to calorie restriction and fasting result in an increase in α-Ketoglutarate, which alters the chromatin structure and DNA accessibility by regulating the activity of KDM and TETs [211] (Figure 3).

FAD^+^, a crucial redox cofactor that plays a vital role in numerous metabolic reactions, operates as a coactivator of histone demethylases (JmjC). Reduced caloric intake, which is associated with caloric restriction and fasting, leads to increased levels of FAD^+^, thereby promoting the demethylation of histones and ultimately resulting in the remodeling of the chromatin structure through the activation of histone demethylases [211] (Figure 3).

Acetyl-CoA, a product of the oxidative metabolism of glycolytic pyruvate, free fatty acids, branched-chain amino acids, and ketone bodies within the tricarboxylic acid cycle, plays a crucial role in epigenetic signaling. It serves as a vital cofactor for HAT acetyltransferase. During periods of fasting or caloric restriction, the scarcity of nutrients leads to an enhancement in autophagy and mitochondrial metabolism, which results in a decrease in metabolic cofactors, including acetyl coenzyme A. This essential compound is crucial for epigenetic remodeling and cell differentiation, leading to a decrease in histone acetylation and subsequent alteration in transcription. Concurrently, a reduction in methionine and its derivatives impacts histone methylation and DNA methylation, resulting in the reorganization of chromatin and DNA [211,212,213] (Figure 3).

Lactylation is a newly recognized epigenetic modification that is dependent on the availability and production of lactate, which is the final product of glycolysis. This modification involves the lactylation of histone lysine residues by lactate, and it acts as an epigenetic modification of chromatin that directly promotes gene transcription. Low-calorie dietary interventions, such as CR, reduce lactate production by shifting the metabolism away from glycolysis towards OXPHOS. Consequently, lactate reduction leads to a decrease in histone lactylation, which ultimately regulates gene expression [214].

Additionally, both low and high glucose levels have an impact on genomic histone GlcNAcylation, which is a critical regulator of cellular processes, such as signal transduction, metabolism, and transcription. As a result, hypoglycemic diets like CR and PF have an effect on gene expression by modulating chromatin GlcNAcylation [215].

Overall, low-calorie diets may have positive effects on aging and age-related diseases linked to inflammation, like cancer, by influencing chromatin remodeling and epigenetic mechanisms, which are regulated by specific diet-dependent metabolites. Thus, the content of protein, carbohydrates, lipids, and calories in a diet can slow the progression of diseases and prevent age-related damage by regulating nutrient sensing pathways, such as IGF-1/PI3K/AKT-mTOR and RAS/PKA, and reshaping the epigenetic landscape (Figure 3).

## 6. Inflammation, Epigenetics, and Cancer

The development of precancerous and cancerous lesions is a complex process that is influenced by a variety of factors, including genetic mutations, epigenetic changes, and environmental stimuli. An unhealthy diet high in saturated fatty acids, glucose, and low in omega-3 fatty acids and fiber can affect the onset and progression of tumors by promoting inflammation and epigenetic remodeling of cancerous cells and cells in the tumor microenvironment, such as immune and stromal cells. This section provides an overview of the relationships between inflammation, epigenetics, nutrition, and cancer.

### 6.1. Inflammation in Tumor Onset and Progression

Chronic inflammation and tumor-induced inflammatory responses play a key role in promoting tumorigenesis and tumor progression by modulating the differentiation of immune and stromal cells towards immunosuppressive subtypes that support immune evasion and tumor growth [4,216]. The release of proinflammatory cytokines, such as IL-1, IL-6, IL-8, and TNF-α, by immune, stromal, and tumor cells activates tumor proliferative and pro-survival signaling pathways, such as NF-κB and STAT3 [217]. Reactive oxygen species (ROS) produced by myeloid cells stimulate the tumor secretion of TNFα, which, in turn, drives the release of proinflammatory cytokines, creating a vicious loop of tumor-promoting factors [218]. The production of ROS and oxidative nitrogen species, such as superoxide, hydroxyl radical, and peroxynitrite, by myeloid cells during inflammation in the tumor microenvironment causes DNA damage. This, in turn, activates the previously mentioned pro-inflammatory pathways, further fueling inflammation. This crosstalk between inflammation and DNA damage creates a feedback loop that promotes tumor progression [219]. Although the initial tumor-induced inflammatory response promotes the recruitment of innate and adaptive immune cells to mount an adequate immune response, an extensive, permanent, and long-lasting inflammatory state suppresses immune function and forces T cells to become exhausted, resulting in tumor immune evasion [220]. Exhausted T cells lose their proliferative potential and effector functions and express immune checkpoint inhibitory molecules, including PD-1, TIM-3, LAG-3, and CTLA-4 [221].

The inflammatory state within the tumor microenvironment could potentially be linked to the dysregulation of the inflammasome as a result of genetic mutations that accumulate within tumor cells. Gain-of-function mutations have been identified in NLRP1 and NLRP3 in various cancer types, including self-healing palmoplantar carcinomas, nodular melanoma, lung adenocarcinoma, small-cell lung cancer, bladder, gastric, and pancreatic cancer. AIM2 and GSDMD are overexpressed in non-squamous non-small-cell lung cancer, while GSDMD downregulation is associated with gastric cancer [222,223,224,225,226,227].

Inflammasomes are activated in tumor cells, tumor-associated macrophages, tumor-associated fibroblasts, and bone-marrow-derived suppressive cells, resulting in the secretion of IL-1β and IL-18 in the TME [228,229,230,231]. The release of IL-1β promotes the migration, invasion, and metastasis of melanoma and gastric cancer cells by increasing the expression of matrix metalloproteases MMP-2 and MMP-9 [232]. The activation of inflammasomes in myeloid cells, cancer-associated fibroblasts, and tumor-associated macrophages has been found to be positively correlated with the metastasis and poor survival rates of patients with breast and lung cancer [231,233,234,235]. Furthermore, studies have shown that the BRCA1 mutation stimulates inflammasome activation and IL-1β secretion via ROS production and cGAS-STING activation, thereby promoting breast cancer metastasis [236].

However inflammasome activation represents a double-edged sword, as it can promote or suppress tumor growth depending on the disease stage and cell type in the microenvironment [237]. Indeed, the inflammasome suppression in genetically engineered mice prone to cancer development leads to controversial outcomes.

For example, ASC knockout prevents the development of spontaneous intestinal-type gastric cancer, tumors induced by the carcinogen 4-nitroquinoline 1-oxide, and metastatic melanoma [238,239,240]. Conversely, a reduction or downregulation in several components of the NLRP3 inflammasome has been implicated in the pathogenesis of hepatocellular carcinoma, chemically induced squamous cell carcinoma, primary melanoma, and colitis-associated cancer [240,241,242,243]. Inhibition of the IL-1R1 signaling pathway has been shown to result in increased incidence and mortality rates of breast cancer in mouse tumor models [244] On the other hand, the overexpression of IL-1β in the stomach has been associated with increased gastric inflammation and cancer [232], and its expression in lung cancer has been found to enhance tumor angiogenesis and growth [245].

These studies indicate that the role of inflammasomes in tumor development can be either pro- or anti-tumor, depending on the genetic and microenvironmental context and the stage of the disease. In the early stages of tumorigenesis, the inactivation of the inflammasome may facilitate immunoevasion and tumor growth, whereas in more advanced stages, its inactivation may counteract tumor growth by reducing chronic inflammation and preventing T-lymphocyte exhaustion. Further investigations are necessary to determine the precise role of inflammasomes in cancer and in which specific stage or context they may serve as a suitable molecular target.

### 6.2. Epigenetic Alterations in Cancer

The onset and progression of cancer are closely associated with changes in the chromatin structure, resulting in distinct gene expression states and channel-specific phenotypic differentiation [246]. Specific genetic mutations can alter the signaling pathway that drives cells to adopt specific genetic and epigenetic states, ultimately conferring a cancer-related phenotype [247].

The process of epigenetic reprogramming in cancer is not solely attributable to mutations in oncogenes or tumor suppressor genes. Instead, it is influenced by alterations in genes that regulate epigenetic modifications. These changes in epigenetic regulation provide tumor cells with phenotypic plasticity and heterogeneity, allowing genetically identical cells to exhibit distinct phenotypes and temporarily modify their expression. These characteristics of tumor cells contribute to drug resistance, facilitate epithelial–mesenchymal transition (EMT), or enable immune evasion [248,249].

Thus, we provide an overview of the most commonly observed epigenetic modifications and their associations with genetic mutations in tumors.

Histone alterations in cancer. Abnormal patterns of histone and DNA modifications are prevalent in numerous tumor types. In human cancer cells, decreased global monoacetylation and trimethylation of histone H4 (H4K16ac and H4K20me3), altered methylation status of H3K9 and H3K27, and hypomethylation of repetitive DNA sequences are often observed [250]. These alterations in the epigenetic landscape result from mutations in genes of histone, epigenetic writers, readers or chromatin remodeler.

Somatic alterations in H3, particularly at amino acids K27, K36, and G34, promote the tumor initiation and progression of pediatric high-grade gliomas, including glioblastomas (GBMs) [251]. These amino acid substitutions affect H3 methylation and acetylation, leading to an abnormal chromatin structure and gene expression.

Altered expression or mutation of epigenetic writers is frequently associated with several tumor types, including melanoma and breast, bladder, endometrial, renal cell, liver, and lung cancers, and it is often involved in tumor progression and metastasis [252].

HATs are crucial in tumor development, as demonstrated by recurrent chromosomal translocations such as MLL-CBP and MOZ-TIF2, coding mutations (e.g., p300/CBP), and altered expression in solid and hematological malignancies [253,254]. In acute myeloid leukemia, the fusion protein from the genetic translocation between MOZ (acetyltransferase) and TIF2 (nuclear receptor coactivator) recruits the HAT protein CBP to the nucleosome, leading to the aberrant acetylation of histones, activation of a self-renewal program, and conferral of stem properties, ultimately contributing to leukemia progression [255].

Somatic mutations in HDACs do not seem to be a significant factor in the development of cancer, yet the expression levels of various HDACs are altered in many types of malignancies. Cancer-related chimeric fusion proteins, such as PML-RARa, PLZF-RARa, and AML1-ETO, have been found to contribute to leukemogenesis by bringing HDACs to inappropriately silence genes, thereby promoting the development of leukemia [256].

EZH2 is a component of the polycomb repressive complex 2 (PRC2), which is responsible for silencing gene transcription through the methylation of H3. Thus, both gain-of-function and loss-of-function mutations in EZH2 can result in chromatin remodeling, which in turn promotes tumor development by either repressing tumor suppressor genes or expressing oncogenes [257]. In epithelioid sarcoma, EZH2 is constitutively active due to the inactivation of the SWI/SNF chromatin remodeling complex, which functions antagonistically to EZH2 by promoting active transcription through chromatin decompaction [258]. The gain of function of EZH2 in mesothelioma is attributed to the mutation of BAP1, a deubiquitinating enzyme and component of the PR-DUB complex. The inactivation of BAP1 results in an increase in ubiquitinated H2AK119ub1, which then recruits the PRC2 complex and facilitates the methylation of H3K27me3 via EZH2. This leads to increased chromatin compaction and gene repression [259]. Different coding mutations within EZH2 have been identified in various lymphoid and myeloid neoplasms. EZH2 mutations found in diffuse large B-cell lymphoma [260] increased the conversion of H3K27me1 to H3K27me2/3, ultimately activating a pro-tumor transcriptional program. In contrast, loss-of-function mutations in EZH2, which confer a poor prognosis, have been described in myeloid malignancies, T-ALL, suggesting a tumor-suppressive role for EZH2 in these cell lineages [261,262].

Cytogenetics and next-generation sequencing analysis of diverse cancer genomes have consistently revealed frequent translocations and/or coding mutations in numerous KMT genes. In acute lymphoblastic leukemia, rearrangements of the histone lysine N-methyltransferase 2A (KMT2A) gene result in the production of aberrant fusion proteins that recruit DOT1-like histone lysine methyltransferase (DOT1L) to incorrect sites and methylates H3K79, causing the expression of oncogenes such as HOXA9 and MEIS1 [263].

Alterations in the expression of the lysine-specific histone demethylase LSD1 are frequently observed in various hematopoietic malignancies and solid tumors, including breast, lung, colorectal, and prostate cancers [264,265,266,267]. LSD1 plays a role in regulating self-renewal and differentiation of hematopoietic and neuronal stem cells, as well as tumor stem cells, by suppressing gene transcription through the demethylation of H3K4me1/2 [268,269].

Histone and DNA reader alterations in cancer. BET proteins have been implicated in a variety of diseases, including cancer, inflammation, and metabolic disorders. These proteins have been found to promote cancer development by altering gene expression in both tumor and TME cells [270]. Specifically, the overexpression of BET proteins has been observed in several types of tumors, such as acute myelogenous leukemia (AML), Burkitt lymphoma (BL), multiple myeloma (MM), prostate cancer (PCa), and breast cancer [271,272].

Multiple studies have reported alterations in the histone methylation lysine reader sequence or expression in both solid and hematological tumors [273]. Nevertheless, the precise mechanisms that connect the disruption of methyl-lysine binding to chromatin to cancer development are not yet fully understood [274].

DNA methyl binding proteins (MBDs) serve as epigenetic readers because they bind to methylated DNA and recruit enzymes that modify histones to coordinate chromatin processes. Mutations in MBD proteins have been identified in various human cancers and have been shown to be critical in the development of neoplasms [275,276,277,278].

Alteration in chromatin remodeler in cancer. Chromatin remodelers are complexes that utilize ATP to shape the chromatin structure by moving, evicting, and exchanging histones. They act in response to the epigenetic modifications of chromatin and DNA generated by various epigenetic writers. There are four main families of chromatin remodelers: switch defective/sucrose non-fermenting (SWI/SNF), imitation switch (ISWI), nucleosome remodeling deacetylase/Mi-2/chromodomain helicase family (NuRD/Mi-2/CHD), and inositol requiring 80 (INO80) family [279]. Chromatin remodeling complexes play a crucial role in granting access to condensed genomic DNA to regulatory transcription, thereby functioning in opposition to polycomb complexes, which repress the chromatin structure. Alterations and mutations in chromatin remodeler complexes can lead to an imbalance in the chromatin structure, which can promote tumor malignancy by disrupting the balance between cell self-renewal and differentiation genes and upregulating the expression of genes involved in cell cycle progression, cell motility, and nuclear hormone signaling [280]. Several members of the SWI/SNF complex have been found to be mutated at high frequencies in hematological tumors and solid malignancies, suggesting that they play a role in the development and maintenance of cancer [281,282,283,284,285,286].

For instance, mutations in ARID1A, a component of the SWI/SNF complex, have been identified in several types of cancer, including endometrial, bladder, gastric, liver, biliopancreatic, and aggressive ovarian cancers. These mutations can cause changes in histone acetylation and methylation [287,288]. Additionally, the overexpression of ARID1A has been linked to hepatocellular carcinoma (HCC) [289].

Aberrant DNA methylation in cancer. Tumors are characterized by abnormal methylation patterns, primarily due to changes in the activity and function of DNA methyltransferases (DNMTs). DNMTs can act as both oncogenes and tumor suppressors, as mutations that activate or inactivate DNMTs can promote tumor development by inducing hypermethylation of tumor suppressor gene CpG islands or by inducing global DNA hypomethylation, which is correlated with greater genomic instability due to the reactivation of RTEs and relaxed chromatin structure [290]. Aberrant expression of DNMTs has been observed in several tumor types, including lymphoma, breast, lung, colon, liver, melanoma, pancreas, prostate, and esophagus [291].

TET dioxygenases are crucial for maintaining DNA methylation levels, as they collaborate with TDG to initiate DNA demethylation. Inactivation of these enzymes leads to the hypermethylation of DNA, resulting in severe disturbances in gene expression and increased incidence of tumor development and embryonic lethality in mice [116,292,293,294]. As a result, alterations in TETs and TDG have the potential to serve as drivers of cancer, even though only aberrations and mutations in TET family genes have been found in tumors. Recurrent MLL chromosomal translocations involving TET1 have been observed in acute myeloid leukemia [295], while recurrent mutations in TET2 have been detected in various hematological malignancies [296,297]. Mutations in the TET family in tumors are all loss-of-function mutations, resulting in an increase in 5mC levels and enhanced self-renewal properties for malignant clones [296].

Although mutations or alterations in TDG have not been identified, TDG and TET genes could potentially be epigenetically silenced, contributing to cancer oncogenesis. Indeed, genomic sequencing of various tumor types has revealed alterations in the methylation promoter regions of several genes, acting as diagnostic markers, even in the absence of mutations or alterations in DNA demethylation genes.

In colorectal cancer (CRC), the hypermethylation of NDRG4, SDC2, and BMP3 promoter regions has been associated with the development of the disease [298]. Additionally, the methylation of BCAT and IKZF1 correlates with CRC stage progression [299]. In cervical precancerous lesions and cervical cancer, the promoter regions of DLX1, ITGA4, RXFP3, SOX17, and ZNF671 are often hypermethylated [300]. The progression of cervical intraepithelial neoplasia (CIN) from low grade (CIN1) to high grade (CIN3) is correlated with the methylation of the promoters of FAM19A4, EPB41L3, JAM3, and PAX1 genes [301].

The methylation of specific genes, including VASH2, CHFR, GRID2IP, CCNJ, SEPT9, and F12, is considered a biomarker for the onset of hepatocellular carcinoma (HCC) [302]. Silencing of the ONECUT2 and VIM genes through promoter methylation has been observed in bladder tumors [303], while promoter hypermethylation of the CCNA1 and VIM genes is associated with esophageal cancer [304]. In prostate cancer, the predominantly methylated genes are GSTP1, APC, and RASSF1 [305], while methylation of the MGMT promoter gene is found in gliomas and serves as a good prognostic marker for response to alkylating agents [306].

miRNA dysregulation in cancer. Alterations in the expression of microRNAs (miRNAs) have been observed in numerous tumors. These small non-coding RNAs can act as either tumor suppressors or oncogenes, depending on the genes they regulate. When miRNAs inhibit the expression of tumor suppressor genes or promote the transcription of oncogenes, they are considered oncogenes. Conversely, when they inhibit the expression of oncogenes, they are classified as tumor suppressors [307]. Thus, the development of several tumors is associated with a reduction in tumor suppressor miRNAs or an overexpression of oncogenic miRNAs. For instance, the expression of miR-15 and miR-16 is reduced in chronic lymphocytic leukemia (CCL) [308], while a low expression of miR-146a, an inhibitor of NF-κB, is linked to the progression of stomach cancer [309]. The depletion of let-7, miR-34, miR-342, miR-345, miR-9, miR-129, and miR-137 promotes the onset and progression of colorectal cancer (CRC) [310]. Breast cancer development is associated with reduced levels of miR125-b, miR-200, miR-205, and miR-145, which regulate apoptosis and epithelial–mesenchymal transition (EMT) [311]. In contrast, the overexpression of several miRNAs promotes tumor onset, progression, and metastasis. For example, miR-10b is overexpressed in triple-negative breast cancer (TNBC), and its expression correlates with tumor metastasis and drug resistance [312]. Several miRNAs, such as miR-17-92, miR-17, miR-18, miR-19a, miR-20, miR-19b, and miR-92, belonging to the oncomiR-1 cluster, repress the transcription of the Rbl2 suppressor gene, and their overexpression promotes tumor cell proliferation and growth [313]. Similarly, high levels of miR-31 and miR-21 stimulate tumor growth by silencing cyclin-dependent kinase (CDK) inhibitors [311].

lncRNAs involvement in cancer. Long non-coding RNAs (lncRNAs) are also implicated in tumorigenesis and cancer progression [314]. MALAT1 is overexpressed in breast, colorectal, gastric, and liver cancers [315], while high levels of HOTAIR are associated with bladder, colorectal, and gastric tumors [316]. Additionally, the overexpression of CCAT2 has been detected in colorectal and esophageal cancers [317]. Generally, aberrant expression of these lncRNAs in tumors correlates with poor prognosis and lower overall survival of patients.

Epigenetic modifications driven by oncogene and tumor suppressor gene mutations. Mutations in oncogenes and tumor suppressors or external stimuli can affect the epigenetic landscape by modulating tumor metabolism, influencing the transcription of genes responsible for epigenetic processes, and controlling the accessibility of proteins, involved in epigenetic modifications, to chromatin and gene promoter regions.

Oncogene mutations can modify the epigenetic landscape by influencing cellular metabolism and releasing metabolites that impede the activity of epigenetic writers. In the context of acute myeloid leukemia (AML) and glioma, oncogenic mutations in isocitrate dehydrogenases IDH1 and IDH2 result in the conversion of α-ketoglutarate to 2-hydroxyglutarate (2HG), a substance that inhibits Jumonji-C domain histone demethylases (JHDMs) and TET dioxygenase, involved in DNA demethylation. This inhibition of histone and DNA demethylases alters the global methylation landscape and accelerates cancer progression [318].

p53 is the most commonly mutated tumor suppressor gene, and its alterations or loss can significantly impact epigenetic reprogramming, as well as rendering tumor cells resistant to apoptosis. Specifically, p53 regulates the expression of DNMT3A and DNMT3B, as well as the transcription of TET1 and TET2 in embryonic stem cells [319]. Thus, the loss of p53 function leads to altered expression of these epigenetic modifiers, resulting in epigenetic heterogeneity and tumor progression [320].

Mutations in genes that regulate tissue-specific differentiation can increase the risk of tumor development by enhancing cell plasticity and epigenetic variability. The deficiency of lineage-specific transcription factors can lead to the recruitment of epigenetic modulators, such as the PcG complex, to promoter regions of target genes and reshape the epigenetic landscape. The PcG complex can create an epigenetic state called bivalency by depositing the repressive H3K27me3 mark in opposition to the activating H3K4me3 mark in the promoter region of target genes. Bivalently marked genes are characteristic of stem cell plasticity, allowing for rapid gene activation or deactivation by losing or maintaining the repressive H3K27me3 mark. These epigenetic and genetic changes could potentially predispose cells to neoplastic transformation. For example, loss of the WT1 tumor suppressor function, involved in renal differentiation, can endow pronephric cells with mesenchymal stem cell properties and the ability to differentiate into mesodermal tissues, such as skeletal muscle, fat, cartilage, and preneoplastic nephrogenic rest cells [321,322].

Environmental stimuli, such as inflammation, can cause changes in the epigenetic landscape, which may increase the risk of tumor development. For example, cytokine release, triggered by reflux-related inflammation, can lead to the expression of CDX1 and CDX2, intestine-specific factors, in Barrett’s esophagus. The CDX1 and CDX2 expression promotes the transdifferentiation of the normal squamous epithelium into the columnar epithelium with intestine-like features, which is a significant risk factor for the development of esophageal adenocarcinoma [323].

### 6.3. Inflammation-Induced Epigenetic Alterations in Tumor Immune Microenvironment

Chronic inflammation can lead to changes in the TME that trigger metabolic, genetic, and epigenetic alterations in both the tumor cells and the TME cells [324]. Epigenetic changes can have a significant impact on gene expression and cell identity, potentially leading to the development of malignant and metastatic tumor cells. In addition to cancer cells, abnormal epigenetic modifications also take place in immune cells present in the tumor microenvironment. These modifications contribute to the creation of an immune-tolerant state by inducing T-cell exhaustion, enhancing the immunosuppressive activity of myeloid-derived suppressor cells (MDSCs) and T-regulatory cells (Tregs) and decreasing tumor-associated antigens and T-cell co-stimulatory signals. Cancer and inflammation in the TME exert their immunosuppressive influence by hijacking the same epigenetic mechanisms involved in the differentiation and activation of immune cells, such as macrophages, T cells, and NK cells [325]. Below, we summarize the epigenetic alterations most commonly found in the TME’s innate and adaptive immune systems.

Epigenetic regulation of innate immune cells. The function of dendritic cells, which are essential for T-cell-mediated immunity, is altered in the TME. The epigenetic changes in the TME impede the maturation of dendritic cells, leading to immunotolerant phenotypes with a low expression of the major histocompatibility complex (MHC, a key component in antigen-presenting machinery) and co-stimulatory molecules [326]. In pancreatic and colon cancer, FOXM1 overexpression, a transcription factor that suppresses DC maturation and function, is due to increased H3K79me2 methylation caused by dysregulated DOT1L methyltransferase activity [327]. In the TME, KLF4 activation in dendritic cells inhibits their maturation by promoting IL-6 release, a dendritic maturation inhibitor, through hyperacetylation of histones in the IL-6 promoter [328,329]. Simultaneously, IL-10 production in the TME suppresses MHC II expression by inhibiting CIITA type I expression, a positive regulator of MHC-II, through the inhibition of STAT5-associated histone acetylation of the CIITA locus [330].

The microenvironment of tumors alters the function of macrophages, which are vital for both innate and adaptive immune responses, by promoting their polarization from a pro-inflammatory M1 phenotype with anti-tumor properties to an anti-inflammatory M2 phenotype with pro-tumor properties. Epigenetic mechanisms play a key role in regulating the polarization of M1/M2 macrophages through the control of their key transcription factors [331]. The release of IL-4 in the TME promotes the expression of histone demethylase JMJD3 in macrophages, which then promotes the transcription of M2 gene markers (such as Arg1 and Retnla) by demethylating dimethyl and trimethyl H3K27 (H3K27me2/3) in their promoter regions [141]. Similarly, IL-4 promotes M2 polarization by inducing the expression of PPAR-γ, a key regulator of M2 macrophages, by inhibiting DNMT3b expression [332]. DNMT3b suppresses the expression of PPAR-γ by methylating the DNA of the promoter region. The activation of PRMT1 arginine methyltransferase promotes M2 polarization by inducing the expression of PPARγ through methylation of H4R3me2a on the PPARγ promoter [333]. The acetylation of histones also plays a crucial role in the polarization of macrophages. Specifically, HDAC3 suppresses the M2-polarized phenotype by repressing many genes regulated by IL-4 [334], while HDAC4 acts as a positive regulator of the M2 phenotype by inducing the expression of Arg1 via STAT6 [335].

Myeloid-derived suppressor cells (MDSCs), including immature neutrophils and monocytes, are negative prognostic markers. They infiltrate the tumor microenvironment (TME) and create an immunosuppressive environment that permits tumor escape. The accumulation and immunosuppression functions of MDSCs in the TME rely on epigenetic processes. The immunosuppressive activity and proliferation of MDSCs depend on the production of immunosuppressive agents Arg1 and S100A8, which are regulated by the activation of STAT3. STAT3 activation relies on DNMT3a and DNMT3b, which suppress STAT3 expression by methylating its promoter region [336]. The inhibitory effects of HDAC11 and HDAC6 on IL-10 (an activator of STAT3) resulted in the suppression of MDSC proliferation [337]. Conversely, CBP/EP300 promotes MDSC expansion by acetylating H3K27 in the promoters of MDSC-associated genes (such as Arg1 and iNOS) [338]. HDAC2 promotes the expansion of immunosuppressive polymorphonucleocytes (PMN-MDSCs) by silencing the retinoblastoma gene (Rb), a transcriptional regulator [339].

The status of natural killer (NK) cells, which play a crucial role in the immune response against tumors by eliminating target cells and secreting cytokines, is often altered in tumors and is typically associated with reduced expression of activating receptors, such as NKG2D, NKp46, and KIR2DS, as well as an increase in the expression of inhibitory receptors, such as NKG2A and TIGIT. This is mainly due to aberrant DNA methylation and histone acetylation–methylation in the promoter regions of these receptors. In patients with hepatocellular carcinoma (HCC), it has been observed that the repression of NKG2D is linked to the hypermethylation and hypoacetylation of histone H3K9Ac in the NKG2D promoter region [340]. Additionally, EZH2 dysfunction leads to the hypermethylation of histone H3K27, which inhibits NKG2D ligand expression [341]. The inactivation of the demethylase KDM5a also impairs NK cell activation by increasing the expression of the suppressor of cytokine signaling 1 (SOCS1) upon accumulation of the active transcription marker H3K4me3 in the promoter region [342].

Epigenetic regulation of adaptive immune cells. CD8^+^ T cells are crucial in limiting tumor growth and preventing its progression. The differentiation of T-naive cells into T-effector cells, central memory cells, effector memory cells, and exhausted T cells depends on epigenetic mechanisms. However, the tumor and tumor microenvironment manipulate these mechanisms to render T cells dysfunctional and exhausted. T-cell exhaustion is characterized by an augmentation of transcriptional activity in genes associated with exhaustion, which is attributed to chromatin remodeling and relaxation. The expression of exhaustion markers, such as Tim-3, HMG box transcription factor Tox, and Tox2, is closely tied to the demethylation of their respective promoter regions [343]. The lack of memory T cells in the tumor immune infiltration is due to specific demethylation at tissue-resident effector gene loci, such as CD39 and CD103 [344]. In pancreatic cancer, tumor-associated macrophages (TAMs) have the potential to impact the epigenetic landscape of tumor-infiltrating lymphocytes (TILs). Specifically, TAMs can promote H3K4me3 methylation of the Il10 promoter in TILs, ultimately leading to the expression of IL10, which inhibits the CD8^+^ T-cell function by decreasing antigen sensitivity [345].

The involvement of microRNAs (miRNAs) in the remodeling of the TME and the generation of an immunosuppressive microenvironment is noteworthy. In melanoma and leukemia, several miRNAs contribute to the exhaustion of T cells, inhibition of the expression of effector cytokines, and induction of the transcription of immune checkpoint genes. For instance, the expression of miR-15a/16 leads to T-cell exhaustion [346]. Tumor cells can also induce T-cell exhaustion by inhibiting the expression of miR-28 and miR-150, which are involved in the production of IFN-γ, IL-2, and TNF-α, as well as the repression of PD-1 in T cells [347,348]. Moreover, the repression of miR-491, induced by tumor cells through the activation of the TGF-β1 pathway, triggers the apoptosis of TILs by repressing the production of effector cytokines such as IFN-γ and the transcription of proliferation and survival factors, such as TCF1 transcription factor, cyclin-dependent kinase 4 (CDK4), and extra-large B-cell lymphoma (BCl-xL) [349].

The immunosuppressive role of Treg cells is a crucial factor in tumor immune evasion. FoxP3^+^ Treg cells play a pivotal role in this process. FoxP3 is a key transcription factor for the immunosuppressive function of Treg cells, and its expression and function are dependent on EZH2 [350]. EZH2 promotes the maintenance of Foxp3, while at the same time suppressing the FoxP3 target genes that act as modulators or stimulators of T-cell function by promoting the methylation of H3K27me3, a repressive transcription marker [351,352].

### 6.4. Nutrition and Cancer

Nutrition and diet have recently received considerable attention because of their potential impact on the development of neoplasms and as a complementary therapy to enhance the effectiveness of anti-tumor treatments. Despite numerous epidemiological studies assessing the effects of different dietary habits on health and cancer incidence, the results remain inconclusive. These studies found statistical differences among various dietary patterns, but the differences were small and difficult to reproduce [353,354,355]. Confounding factors, such as socioeconomic status, age, physical activity, food quality, and cooking methods, may also influence the observed benefits of different dietary habits.

Nutrition can affect tumorigenesis in several ways, including modulating the immune system and inflammatory state; regulating endocrine factors, such as circulating insulin, insulin-like growth factor, leptin, and adiponectin [356]; and influencing the gut microbiota [357]. Although the mechanisms by which dietary macronutrients and micronutrients affect the inflammatory state are not yet fully understood, vitamin deficiencies, excessive food intake, and food deprivation have been shown to modulate the immune system and inflammatory processes [358].

Overnutrition refers to the intake of excessive amounts of both macro- and micronutrients, which are then stored in the body’s tissues, especially adipose tissue. When there is no longer any space to store these excess nutrients, adipocytes swell and undergo changes that result in chronic inflammation, which can lead to noncommunicable diseases, such as diabetes mellitus, coronary artery disease, and stroke. Overnutrition is considered a form of malnutrition due to its detrimental impact on health. It is distinct from a high-calorie diet, which is often linked to fast food and is rich in macronutrients but poor in micronutrients, resulting in a low nutritional value.

The overconsumption of food can lead to obesity, hyperlipidemia, and metabolic syndrome, which can cause chronic inflammation and predispose one to cancer [359]. These conditions are associated with an increase in pro-inflammatory macrophage M1 and a decrease in anti-inflammatory M2 macrophages in adipose tissue, as well as an imbalance in the intestinal microbiota that leads to an accumulation of bacteria that produce procarcinogenic metabolites [356,360]. Additionally, free fatty acids can activate Toll-like receptors, leading to the activation of the pro-inflammatory NF-κB and JNK1 pathways [361]. Overnutrition also leads to an increase in acetyl-CoA, overactivation of mTOR, and reduction in the autophagic protein ATG7, all of which cooperate in blocking autophagy [362]. The inhibition of autophagy can lead to the accumulation of damaged organelles that favor the aging process, activation of the NLRP3 inflammasome [363], and accumulation of oncogenic P62 [364]. Furthermore, a high-fat diet can promote intestinal inflammation through the production of the complement C5a fragment [365]. 

Overnutrition impaired the immune system in obese individuals by reducing CD8^+^ cytotoxic T cells, circulating Vγ9^+^Vδ2^+^ T cells, and dysregulating dendritic cells and natural killer cell function [366,367,368].

An unbalanced diet (excess or deficiency of macro- or micronutrients) may increase the risk of developing metabolic syndrome, which is characterized by elevated levels of various markers, such as C-reactive protein, glucose, IL-6, insulin, leptin, triglycerides, and low adiponectin. These changes can lead to inflammation and impair the immune system, increasing the risk of certain types of cancer. For example, obese individuals with metabolic syndrome are at a higher risk of developing breast cancer, which is linked to inflammation in white adipose tissue, as indicated by the presence of macrophages arranged in crown-like structures [359]. A calorie-rich diet leads to a persistent state of inflammation, as elevated blood glucose levels and oxidative stress result in the production of Advanced Glycosylated End products (AGEs). Specifically, AGEs such as glycated proteins bind to the receptor for advanced glycation end products (RAGEs), which in turn stimulates the production of pro-inflammatory cytokines by activating NF-kB [369].

Micronutrients, such as vitamin B6 and 25-hydroxyvitamin D, have been found to exhibit anti-tumor activity and to enhance anti-tumor immunosurveillance by preventing DNA damage and inflammation. High consumption of vitamin B6 and elevated plasma concentrations of its metabolite, pyridoxal-5′-phosphate (PLP), have been linked to a reduced risk of cancer, including gastrointestinal tumors [370]. Increased levels of 25-hydroxyvitamin D in serum have also been associated with a decrease in pro-inflammatory markers and favorable prognosis in patients with breast cancer, prostate cancer, or colorectal cancer [371,372,373]. Therefore, deficiencies in these vitamins may be risk factors for various types of cancer [358].

Folate, also known as vitamin B9, is crucial for methylation by providing methyl donors. Adequate folate intake may reduce the risk of developing certain types of cancer, such as esophageal squamous cell cancer, breast cancer, pancreatic cancer, and cervical intraepithelial neoplasia [374,375,376]. However, excessive folate intake may increase the risk of skin cancer, particularly basal cell carcinoma and non-melanoma skin cancer, especially in women [377]. The beneficial effects of folate in preventing cancer may be related to its ability to reduce inflammatory markers, while the detrimental effects of overconsumption may be related to epigenetic alterations in DNA and histone methylation.

The impact of diet on cancer risk may also be mediated by altering the composition of the microbiota, which has been shown to influence the response to immunotherapy. Notably, studies have revealed that the microbiota can bolster the anti-tumor immune response by releasing microorganism-specific peptides. These microbial-derived peptides share similarities with tumor neoantigens and can reinvigorate the anti-tumor immune response by directly priming CD8^+^ T cells or through the activation of the innate immune system [378,379].

In general, inflammation is a critical factor in the development and progression of cancer, and it can be triggered by external factors, such as pathogens, diet, and lifestyle, as well as by internal factors induced and modulated by the cancer itself. The tumor genomic instability and gene mutations can initiate a sterile inflammatory response and amplify it by recruiting myeloid cells to the tumor microenvironment and by inactivating and exhausting cytotoxic T cells. This inflammatory state increases tumor DNA instability and epigenetically reprograms both tumor and immune cells to promote a pro-inflammatory response, which in turn drives the ongoing cycle of inflammation and epigenetic modifications. On the other hand, diets, depending on their components and calorie intake, can have either beneficial or detrimental effects on both inflammation and cancer by controlling the signaling pathways and epigenetic mechanisms that are connected to inflammation and cancer. Understanding the epigenetic mechanisms and signaling pathways that regulate this vicious circle is crucial in order to develop new pharmacological and dietary interventions to improve the life expectancy of cancer patients.

## 7. Cancer Therapies and Future Directions

### 7.1. Anti-Inflammatory Drugs

Nonsteroidal anti-inflammatory drugs (NSAIDs) may offer a valuable strategy for combating inflammatory cancer by targeting chronic inflammation. NSAIDs work by inhibiting prostaglandin through the COX-1 and COX-2 enzymes, with different types of NSAIDs having varying levels of selectivity [380]. COX-1 inhibitors include flurbiprofen, ketoprofen, fenoprofen, oxaprozin, and tolmetin, while non-selective NSAIDs include indomethacin, ibuprofen, naproxen, piroxicam, ketorolac, and nabumetone. COX-2 inhibitors are further divided into two classes: (1) the old class includes sulindac, meloxicam, salsalate, etodolac, mefenamic acid, diclofenac, nimesulide; (2) the new class encompasses celecoxib, valdecoxib, rofecoxib, etoricoxib, and lumiracoxib [381].

Indomethacin, celecoxib, and other nonsteroidal anti-inflammatory drugs (NSAIDs) have demonstrated promising results in impeding the growth, migration, and invasion of various cancer cells by suppressing COX-2 expression and inducing cancer cell apoptosis [382,383]. Some NSAIDs inhibit PI3K/Akt signaling, leading to cell arrest in the G0-G1 phase. Additionally, celecoxib and acetylsalicylic acid suppress Wnt/β-catenin signaling, resulting in a reduction in migration and cell viability in colon cancer cells [384]. Celecoxib inhibits the survival of adenocarcinoma gastric cells by inducing the expression of the wildtype p53 [385]. Acetylsalicylic acid reduces nuclear factor-kB (NFkB) levels in human hepatocellular carcinoma cells, leading to decreased abnormal lipid metabolism [386]. Furthermore, diclofenac inhibits leukemia cell proliferation by downregulating c-MYC gene expression [387].

NSAID use has also been associated with a decreased risk of stomach, large bowel, and stomach cancer and chemopreventive potential in colorectal cancer [388]. However, NSAID intake is associated with a higher risk of stroke and with upper gastrointestinal complications, such as peptic cancer cases and acute kidney injury [389]. Rofecoxib is the most highly associated with cardiovascular risk, whereas ibuprofen is associated with a higher risk of stroke [389].

NSAIDs in combination with chemotherapy affect cancer progression significantly. In a phase two trial with advanced solid tumor patients, a combination of rofecoxib, cyclophosphamide, and vinblastine showed a 30% clinical benefit response [390]. In several studies, the administration of celecoxib in combination with carboplatin had a low impact on anti-angiogenic and anti-metastatic features [391]. However, celecoxib with cyclophosphamide decreased serum VEGF levels, indicating anti-angiogenesis potential in metastatic breast cancer [392]. The present findings are consistent with preclinical evidence demonstrating that administering celecoxib orally enhanced the responsiveness of patients who were resistant to platinum-based treatments [393].

### 7.2. DNA Damage Response-Targeted Therapy

The potential therapeutic benefits of DDR inhibitors may include initiating local inflammation, which could promote immunosurveillance and enhance the efficacy of immunotherapeutic agents such as immune checkpoint inhibitors (ICIs) [62]. Preclinical findings suggest that the use of suboptimal doses of DNA-damaging agents in combination with ICIs or other immunotherapeutics can effectively stimulate tumor-targeting immune responses by increasing the inflammatory response via cGAS activation [394].

Inhibitors of PARP, nuclear proteins involved in the cellular response to DNA damage, have been shown to change the TME to a more inflamed state [395]. The production of IFN-I, which leads to the secretion of cytokines, is accomplished through the activation of the cGAS pathway in response to the accumulation of cytosolic single-stranded DNA (ssDNA) or micronucleation [92,396]. PARP inhibitors enhance CD8^+^ T effector cell (Teff cell) recruitment and priming by secreting CCL5 and CXCL10 while simultaneously reducing the immunosuppressive activity of MDSCs [91,397]. PARP inhibitors upregulate the expression of immunosuppressive molecules, such as PD-L1, not only on cancer cells but also on TAMs, and augment the therapeutic benefits of immune checkpoint inhibitors in preclinical studies across a range of tumor types [398]. Combining PD-1 blockers with PARP inhibitors has shown promising results in patients with refractory TNBC or ovarian carcinoma [399,400]. Similar findings have been observed in high-risk patients with HER2- breast cancer who received olaparib plus a PD-L1 inhibitor and neoadjuvant chemotherapy [401].

ATM and WEE1 (DNA damage cell-cycle checkpoint protein) inhibition has been shown to decrease CXCR2 expression and surface PDL1 exposure in human pancreatic ductal adenocarcinoma (PDAC) cells with KRAS mutations [402]. The suppression of ATM through genetic and pharmacological means has been demonstrated to trigger intense type I interferon signaling in breast and melanoma cancer cell lines [403]. This results in the production of CCL5 and CXCL10, the attraction of CD8^+^ CTLs, and a response to ICIs in vivo [404]. The use of ATR inhibitors in radiotherapy treatment can lead to an increase in the secretion of IFN-I, CCL5, and CXCL10. This occurs as a result of the accumulation of cytosolic double-stranded DNA and the subsequent activation of cGAS signaling [405]. In BRCA2-mutant breast cancer, ATR inhibitors in combination with olaparib can also boost antigen presentation and inflammation in the TME by abrogating PDL1 upregulation and Treg cell recruitment [406]. ATR inhibition can also inflame the TME in the absence of DNA-damaging agents, such as through the direct activation of a CDK1 axis, leading to SPOP-dependent PDL1 degradation in prostate cancer cell lines. Similar findings have been observed in preclinical models of small-cell lung carcinoma (SCLC) treated with PDL1 antagonists in combination with the CHK1 (mediator of DNA damage response) inhibitor prexasertib [407].

The exploration of the feasibility of employing DDR inhibitors in conjunction with exogenously administered PRR agonists, such as STING1 agonists or oncolytic viruses, constitutes a promising avenue of investigation. Although the potential of DDR inhibitors as complementary agents for ICIs has been exhibited, the precise influence of DDR-mediated inflammatory reactions on patient outcomes and the likelihood of secondary oncogenesis has yet to be definitively established through formal assessment.

### 7.3. Epigenetic-Targeted Therapy

Recent cancer research has focused on the potential of combination therapies involving epigenetic drugs and immunotherapies, such as inhibitors targeting chromatin remodeling proteins like DNMTs and HDACs. Studies have shown that using HDAC inhibitors like belinostat and trichostatin A (TSA) in combination with anti-CTLA-4 therapy can reduce the number of Tregs and increase the production of IFN-γ by tumor-reactive CD8^+^ T cells in the TME [408,409].

A study on mice showed that combining vaccines for breast and colorectal cancer with entinostat, an HDAC inhibitor, resulted in reduced tumor growth, a more pro-inflammatory microenvironment, and increased type 1 effector immune cells [410].

Preliminary data indicate a synergistic effect when inhibiting DNMTs with checkpoint inhibitors. Decitabine treatment in an OT-I TCR transgenic mouse model showed increased CD8^+^ TILs and reduced tumor growth. Additionally, there was a rise in cytolytic CD8^+^ effector T cells [411].

Different demethylating agents, like DNMT inhibitors decitabine and azacytidine, are used to treat subtypes of leukemia. However, their clinical effectiveness is limited, and the effects cannot be solely attributed to these epigenetic drugs. The restricted drug effectiveness may be due to various factors, such as tumor immune escape mechanisms and exhausted T-cell phenotypes, or it could be because the drug is scarce in the targeted cells [412,413]. Combination therapy with DNMT inhibitors has shown promising results in melanoma patients, overcoming resistance to immunotherapies such as PD-1. The synergistic effects may be due to their impact on highly proliferative cells, including cancer and immune cells. These findings suggest that combining epigenetic therapeutics with cellular immunotherapies may benefit human cancer conditions [414]. The necessary genes and epigenetic mechanisms for fully functional effector T cells or reawakening exhausted T cells are unclear, limiting available druggable targets. Preliminary cancer research shows potential, but further investigation is needed. Indeed, it is important to recognize that various epigenetic drugs have been found to produce unfavorable side effects, such as neutropenia, thrombocytopenia, fatigue, and anemia [415]. These negative consequences have impeded their widespread use in clinical settings.

Pan-BET inhibitors have demonstrated efficacy in preclinical cancer and inflammation models and are currently being studied in clinical trials for various types of cancer treatment. They prevent resistance to targeted cancer therapies by blocking alternative signaling pathways that bypass the drug targets. In TNBC cells treated with lapatinib, pan-BET inhibition with JQ1 or I-BET151 suppresses the transcription of kinases involved in resistance, leading to a durable response [416]. Similarly, in BRAF-mutant melanoma models treated with BRAF/MEK inhibitors, the next-generation pan-BET inhibitor PLX51107 suppresses adaptive receptor tyrosine kinase upregulation in response to targeted therapy. Although the combination of BRAFi/MEKi/BETi is highly toxic, intermittent use of the BETi with continuous BRAFi/MEKi treatment was found to be tolerable and improved durable tumor inhibition outcomes [417].

BET inhibitors can be used to sensitize homologous recombination-proficient cancers to PARP inhibitors. This approach can resensitize acquired PARPi resistance. In preclinical models of breast and ovarian cancers, strong synergy was observed between PARP inhibition and BET inhibition, which prompted the assessment of clinical benefits in patients [418].

Multiple BET bromodomain inhibitors have been tested in clinical trials for hematologic malignancies and solid tumors. Approximately 50 trials have been completed or are in phase 1.Clinical studies on BET inhibitors have validated BRD4 as a drug target for cancer and other indications. However, published data have also revealed dose-limiting toxicity and common side effects, including thrombocytopenia, diarrhea, fatigue, vomiting, anemia, gastrointestinal bleeding, and hyperbilirubinemia [419].

Recent research indicates that BET-BD1 or BET-BD2 selective inhibitors may offer advantages over pan-BET BrD inhibitors in treating certain cancers, inflammation, metabolic diseases, and fibrosis. These selective inhibitors may have reduced adverse toxicity and better tolerability while maintaining anti-tumor activity, making them promising drug candidates for certain diseases [420].

A promising therapeutic approach is inhibiting BrDs and other targets like HDACs and kinases with dual-target inhibitors [421]. However, developing drugs with multiple protein targets for specific diseases is challenging. Dual-target inhibitors could offer advantages over combination therapy by addressing issues, like pharmacokinetic differences, off-target toxicities, drug–drug interactions, and additive effects.

### 7.4. Dietary Interventions as Adjuvant Therapy

Dietary interventions can significantly influence the metabolism, tumor growth, and therapeutic responses by modulating the inflammatory state, signaling oncogenic pathways, and epigenetic mechanisms. Preclinical and early clinical studies suggest that specific dietary patterns can prevent and treat cancer by delaying tumor growth, preventing tumorigenesis, and enhancing the effects of various anticancer treatments [422,423]. Recent studies have revealed a surprising finding regarding the consumption of red meat. Despite the prevailing negative opinion, dietary trans-vaccenic acid (TVA), which is primarily found in ruminant-derived foods, such as beef, lamb, and dairy products, have been shown to enhance the function of CD8^+^ T cells and improve immune therapy by inactivating the immunomodulatory G protein-coupled receptor GPR43 [424].

Dietary interventions in cancer focus on nutrient-restricted approaches, like CR, fasting, and glucose restriction, which provide simple guidelines while maximizing benefits.

The beneficial anti-tumor effects of CR and PF are related to the physiological changes that they induce at a systemic level, in which they decrease glucose and amino acid levels, growth hormone (GH), and insulin growth factor (IGF1) and increase the production of ketone bodies from fatty acids released following lipolysis of adipose tissue. These changes result in the inhibition of the glucose-dependent PKA/RAS protumor pathway, the IGFI- and amino acid-dependent IGFI/PI3K/AKT/mTOR pathway, activation of the AMP/ATP ratio-dependent AMPK pathway, stimulation of autophagy, and a shift in cellular metabolism from glycolysis to oxidative phosphorylation, with subsequent epigenetic reprogramming associated with changes in metabolic intermediates that act as coactivators of epigenetic enzymes [4,198,425,426,427,428].

Glycolysis-dependent cancer cells, such as ovarian, breast, and thyroid cancers, are unable to effectively adapt to the metabolic shift caused by fasting-induced low glucose levels because they are not able to effectively process the fatty acids that accumulate in lipid droplets [196]. CR slows the proliferation of cells by decreasing the activity of the enzyme stearoyl-CoA desaturase (SCD), which is essential for maintaining the fluidity of the cytoplasmic membrane by producing monounsaturated fatty acids (MUFAs) [429].

Indeed, recent research has demonstrated that PF can sensitize cancer stem cells to the glucose analog (2-DG) and increase the effectiveness of cyclin kinase/hormone inhibitor combined therapy, PI3 kinase inhibitors, and metformin against ER^+^, TNBC, and colorectal tumors, respectively [195,430,431].

CR and PF have an anti-tumor effect in acute lymphocytic leukemia (ALL) by decreasing leptin levels. This decrease in leptin may lead to an increase in the leptin receptor (LEPR), which in turn promotes the differentiation of ALL cells by upregulating the PRDM1 tumor suppressor gene, which plays a role in the differentiation of T and B cells [432].

Moreover, fasting may sensitize tumor cells to chemotherapy and pro-oxidant agents by inducing the production of ROS and oxidative stress. PF promotes this effect by diverting metabolic pathways from glycolysis to serine biosynthesis upon serine reduction and altering iron metabolism, thereby reducing the levels of heme oxygenase (HO-1) and ferritin [433,434].

On the other hand, CR and PF may enhance the efficacy of chemotherapy in RAS-mutated colorectal cancer by constraining methionine, a crucial component of nucleotide metabolism, glutathione synthesis, and DNA and histone methylation [435].

PF can improve the anti-tumor response against various cancer cell types by increasing the percentage of CD8^+^ cytotoxic T cells, memory T cells, and stem-like memory T cells, while reducing IGF1 levels and repressing Tregs by modulating IGF1 levels, epigenetic reprogramming, and the production of ketone bodies [196,197,199]. By promoting autophagy and reducing the expression of IGF-1 and HO-1, potent immunomodulators involved in the accumulation of regulatory T cells (Tregs) and the suppression of CD8^+^ cytotoxic T cells, fasting decreases the number of Treg cells, and, consequently, it promotes the proliferation and activation of CD8^+^ T cells in the TME [436,437].

CR and PF have been found to sensitize low immunogenic non-small-cell lung cancer to anti-PD1 immunotherapy, leading to complete tumor remission in preclinical studies. This is achieved by reversing immune evasion through downregulation of the IGF-I signaling pathway [437]. Furthermore, PF cycles have been observed to increase early exhausted effector T cells and decrease late exhausted effector T cells in breast cancer mouse models. These early exhausted effector T cells are crucial for successful immune checkpoint inhibition and have improved patient survival because the immune checkpoint blockade stimulates the expansion and differentiation of early exhausted effector T cells, maintaining long-term T-cell responses [196].

Fasting and caloric restriction may enhance the anti-tumor immune response by counteracting the immunosuppressive environment within the tumor microenvironment. This is achieved by suppressing the RAGE and IGF1 inflammatory pathways [438]. In mice with melanoma and lung cancer who underwent FMD cycles, the levels of inflammatory markers, including NLRP3 inflammasome and leukotriene, were observed to be decreased in both peripheral blood and the heart [4,439].

Recent studies have suggested that CR and PF can influence the anti-tumor immune response by altering the composition of the gut microbiome [440]. The efficacy of chemotherapy and immunotherapy may depend on certain bacterial species present in the gut microbiome that can produce short-chain fatty acids (SCFAs) such as acetic, propionic, and butyric acids [441,442]. In particular, butyric acid has been shown to inhibit histone deacetylases (HDACs), prevent CD8 exhaustion, and promote effector CD8 activation by increasing IL12Rb expression through upregulation of the transcriptional regulator ID2 [443,444].

In the future, it will be fascinating to explore the potential anti-tumor response and collateral effects that may result from the combination of low-calorie diets with anti-inflammatory drugs and epigenetic drugs. According to recent research, promising results have emerged from a study examining the potential benefits of combining calorie restriction with the use of histone demethylase LSD1 inhibitors to treat leukemia in mice.

Specifically, the combination of CR and LSD1 inhibitors was found to completely eliminate leukemia in mice [445].

Undoubtedly, the preclinical studies on the potential of CR and PF in enhancing the effectiveness of anti-tumor therapies and limiting side effects have garnered significant attention. However, it is important to emphasize that conclusions should not be drawn until the results of the numerous trials that have commenced in recent years are available. It is essential to acknowledge that the metabolism of animals varies significantly from that of humans, which could impact the relevance of these findings [446,447,448]. Additionally, the micronutrient content in these diets is currently unknown, and it is unclear what role it may play in the benefits observed in these studies. Lastly, food–drug interactions could also have an impact on the stability and effectiveness of a given therapy, which should be taken into consideration [448]. Although effective anticancer effects are often inconsistent, current and upcoming clinical trials are essential for evaluating the efficacy of dietary interventions alongside established therapies [6,204].

However, smaller trials also explore other interventions, like essential amino acid restriction, Mediterranean diet-based ketogenic diet (KD) variants, and dietary supplements, expanding the potential therapeutic options [423].

Considering the impact of dietary interventions on the protumor signaling pathway, epigenetics, and cellular metabolism, it would be beneficial to determine the genetic, epigenetic, and metabolic abnormalities of each type of tumor. This information could then be used to develop personalized diets that are tailored to each tumor and combined with therapies that target the altered pathways (precision medicine) [6]. The goal of this approach would optimize the efficacy of cancer treatment by taking into account the unique characteristics of each individual tumor.

## 8. Conclusions

Tumorigenesis is a multifaceted process that involves the interplay of genetic, epigenetic, and inflammatory factors. Conventional single therapy has often proven ineffective, as tumors develop alternative means to survive and proliferate. Therefore, adopting a multi-targeted approach that blocks various alternative pathways adopted by the tumor to evade immune response and acquire resistance is essential. However, this approach is limited by the toxicity and adverse effects associated with multiple drugs.

Dietary interventions offer a promising strategy to prevent tumor onset and enhance the effectiveness of anti-tumor therapies while minimizing side effects. Further investigation is required into the nutraceutical-regulated mechanisms that govern tumorigenesis to develop specific dietary interventions for different types of tumors and cancer patients. This will enable the development of targeted dietary interventions that enhance anti-tumor responses while minimizing adverse effects.

## Figures and Tables

**Figure 1 ijms-25-02750-f001:**
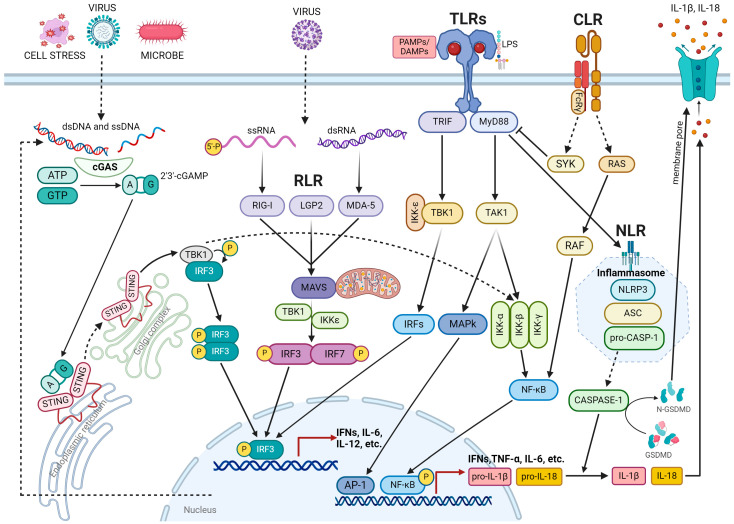
Inflammatory signaling pathways. cGAS is activated upon the recognition of cytosolic double-stranded DNA derived from viruses, bacteria, dead cells, or mislocalized endogenous DNA. Activated cGAS synthesizes 2′-3′ cyclic GMP-AMP (cGAMP), which promotes the translocation of stimulator of interferon gene (STING) from the endoplasmic reticulum (ER) membrane to the ER-Golgi intermediate and Golgi compartments. STING forms a complex with TANK-binding kinase 1 (TBK1), which recruits and activates interferon regulatory factor 3 (IRF3), leading to the transcription of genes encoding inflammatory cytokines such as interleukin 6 (IL-6), interleukin 12 (IL-12), and interferons (IFNs). RIG-I-like receptors (RLRs), including melanoma differentiation-associated gene 5 (MDA5), laboratory of genetics and physiology 2 (LGP2), and retinoic acid-inducible gene I (RIG-I), interact with viral double-stranded RNA (dsRNA) and 5′-triphosphate single-stranded RNA and bind to mitochondrial antiviral-signaling protein (MAVS), which activates IRF3 and 7 via TBK1 and IκB Kinase ε (IKKε). Phospho-IRF3 and phospho-IRF7 transcribe genes encoding IFNs and immunoregulatory genes. Toll-like receptors (TLRs) recruit adapter protein myeloid differentiation primary response 88 (MyD88) and TIR domain-containing adaptor protein (TRIF) upon recognition of pathogen-associated molecular patterns (PAMPs) or damage-associated molecular patterns (DAMPs) on the plasma membrane or in endosomes (not depicted in the caption). MyD88 and TRIF initiate a signaling cascade resulting in the activation of NLR inflammasome, nuclear factor kappa B (NF-κB), interferon regulatory factors (IRFs), or mitogen-activated protein kinases (MAPK) which result in transcription of proinflammatory cytokines and inflammasome component NLR family pyrin domain containing 3 (NLRP3). The stimulation of NLRP3 and apoptosis-associated speck-like protein containing a CARD (ASC) primes the assembly of the inflammasome complex, which triggers caspase-1 mediated cleavage of pro-IL-1β and pro-IL-18 and Gasdermin D precursor, which forms a pore on the plasma membrane through IL-1β and IL-18 being released into the extracellular matrix. C-type lectin receptors (CLRs) play a crucial role in modulating Toll-like receptor (TLR) signaling. They achieve this through two distinct mechanisms: either by activating NF-kB via RAS-RAF1 dependent signaling, or by recruiting spleen tyrosine kinase (SYK) to the phosphorylated immunoreceptor tyrosine-based activation motif (ITAM) of the paired signaling adaptor Fc receptor γ-chain (FcRγ). The recruitment of SYK to FcRγ inhibits the recruitment of MYD88, thereby reducing the production of TLR-induced cytokines (created with BioRender.com).

**Figure 2 ijms-25-02750-f002:**
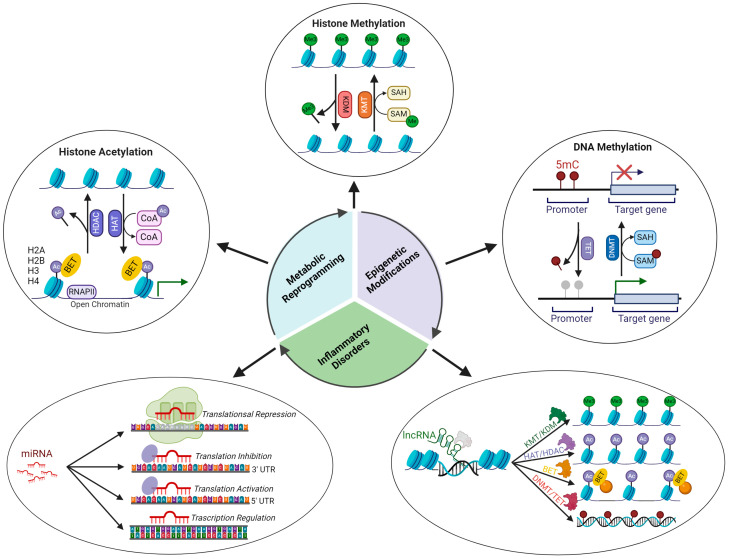
Key epigenetic mechanisms controlling cancer, inflammation, and nutrition. Epigenetic mechanisms such as DNA methylation, histone modifications, and non-coding RNAs play a significant role in regulating the expression of genes associated with inflammation, cancer, and nutrient conditions. In cancer cells, alterations in metabolic pathways occur during tumorigenesis and progression. These metabolic changes are often closely linked to epigenetic modifications and are influenced by inflammation and nutrition. The balance of histone and DNA modifications is crucial for proper regulation. The level of histone acetylation is regulated by the opposing activities of histone acetyltransferases (HATs) and histone deacetylases (HDACs). High levels of histone acetylation promote gene transcription by opening chromatin and facilitating the recruitment of transcription factors and the transcriptional machinery. Histone methyltransferase (KMT) enzymes possess the ability to monomethylate, dimethylate, or trimethylate histone tail lysine and arginine residues, which can subsequently be removed by histone demethylase (KDM) enzymes. The methylation of histones at specific lysine and arginine residues is critical for determining the structure of chromatin and the recruitment of transcriptional repressors or activators. The enzyme DNA methyltransferases (DNMTs) catalyze the DNA methylation of the 5-carbon of cytosine, resulting in the formation of 5-methylcytosine. On the other hand, the removal of the methyl group is carried out by the activity of the ten-eleven translocation (TET) methylcytosine dioxygenase enzymes, which progressively oxidize 5-methylcytosine to 5-hydroxymethylcytosine, 5-formylcytosine, and 5-carboxylcytosine. Thymine-DNA glycosylase (TDG) removes 5-formylcytosine and 5-carboxylcytosine, which are then replaced by cytosine. In human cancers, CpG methylation promotes transcriptional silencing and malignant transformation, while hypomethylation of the transposable element DNA leads to genomic instability and an inflammatory response. Long non-coding RNAs (lncRNAs) can interact with chromatin modifiers and recruit them to the promoters of target genes, where they can activate or suppress transcription. Additionally, lncRNAs can sequester chromatin modifiers away from the promoters of target genes and regulate transcription. Mature microRNAs (miRNAs) are incorporated into a large protein complex known as RNA-induced silencing complex (RISC). This complex either cleaves messenger RNA (mRNA) or induces translational repression by binding to the 3′ untranslated region (UTR) of the target mRNA. Alternatively, it can induce translational activation if it binds to open reading frame (ORF) sequences or the 5′-UTR. miRNAs can modulate gene transcription through direct binding or by altering methylation patterns at the promoter level of the target gene (created with BioRender.com).

**Figure 3 ijms-25-02750-f003:**
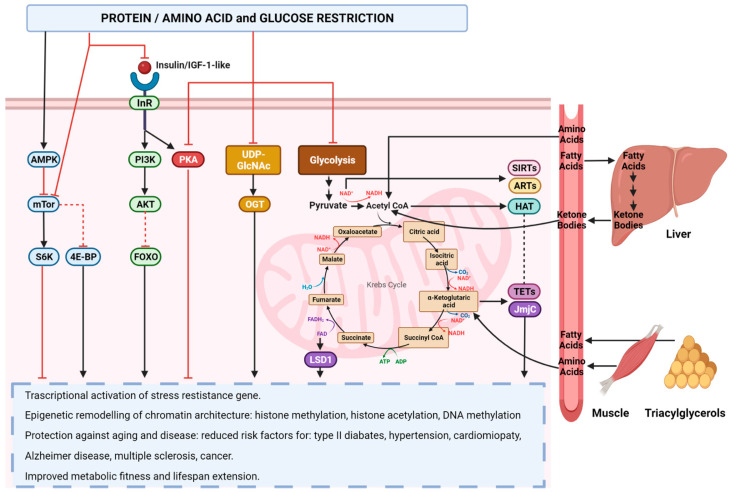
Nutrient-induced metabolic and epigenetic changes. The regulation of cellular genetics and epigenetics by calorie restriction (CR) and periodic fasting (PF) is achieved through the modulation of metabolism and hormonal systems. By reducing the levels of tumor growth-promoting nutrients and factors, including glucose, insulin like growth factor 1 (IGF1), and insulin, these interventions inhibit the IGF1-PI3K-AKT-mTOR and protein kinase A (PKA) signaling pathways, activate AMP-activated protein kinase (AMPK), and lead to the activation of stress-resistance genes that protect against the onset of age-related diseases. The decreased availability of glucose results in the liberation of amino acids and fatty acids from muscle and adipose tissue, respectively. Amino acids are utilized for glucose production through the process of gluconeogenesis, while fatty acids in the liver are converted into ketone bodies, which serve as the primary source of energy during periods of nutritional restriction. These metabolic adjustments shift cellular metabolism from glycolysis to oxidative phosphorylation. These modifications result in the formation of acetyl-CoA, which serves as an acetyl group donor for histone acetyltransferase (HAT)-dependent acetylation of nucleosomal histones. The elevation of NAD+ resulting from energy scarcity or the augmentation of de novo synthesis from amino acids, such as tryptophan, triggers the activation of histone deacetylase (HDAC) sirtuins (SIRTs). These sirtuins subsequently deacetylate histone H3, which in turn modulates the expression of metabolic genes and pathways, including glycolysis, gluconeogenesis, mitochondrial respiration, fatty acid oxidation, and lipogenesis. α-ketoglutarate (α-KG), which is produced from glutamine and participates in the TCA cycle, functions as a cofactor for the catalytic activity of lysine demethylase (JMJc) and ten-eleven translocation (TETs) enzymes, which play a role in DNA demethylation. Variations in glutamine and glucosamine glucose levels have an impact on the biosynthesis of uridine diphosphate N-acetylglucosamine (UDP-GlcNAc) and, consequently, the O-GlcNAcylation (OGT) of histones and proteins, which subsequently influence epigenetic reshaping. FAD+ levels generated by the KREBs cycle modulate the activity of LSD1 histone demethylases (created with BioRender.com).

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
