# Peer review of "The Killer’s Web: Interconnection between Inflammation, Epigenetics and Nutrition in Cancer"

_ijms, 2024, doi:10.3390/ijms25052750_

Round 1

Reviewer 1 Report

Comments and Suggestions for Authors

Author Response

Dear Reviewer,

I would like to express my gratitude for the valuable feedback and recommendations provided to enhance the quality and clarity of our paper. Consequently, we have implemented your suggestions by introducing subheadings for each paragraph and incorporating concluding notes.

We have added the mechanisms regulated by diet in genetic reprogramming, the onset of cancer, and the improvement of therapy effectiveness. We have also restructured the review to better reflect the interconnection between inflammation, nutrition, and cancer. To facilitate identification of the modified sections, we have highlighted them in red.

Thank you for taking the time to review our work, and we hope that the revised version of our paper is now suitable for publication.

Best regards

Please find our responses to each point raised below.

 P2 line 46 – s is missing

We have corrected this typo

 P2 Line 65: It is Phosphorylated IRF3 and IRF7 that act as transcription factors.

We added phospho to IRF3 and IRF7

 P2 line66: can be on the plasma membrane or in endosomes. You can indicate that endosomes are not shown in the caption.

We added this note in the figure legend

 P3 line 105: dozens of genes are induced. This should be made clear.

We added dozens in the text

P4 line 150: it would be good to define “pyroptosis”

We have added this definition of pyropotosis: “a form of programmed inflammatory cell death dependent on the activation of inflammasomes and initiated by microbial infections  or various pathological stimuli, such as stroke, heart attack, or cancer”

 P4 line 152: It is not clear how LPS leads to activation of caspase 4 and 5. How does LPS get into the cytoplasm? Up until this paragraph, the organization works well. For the paragaphs starting on line 152 and 167, it is less clear how these relate to the text preceding these paragraphs. Perhaps topic sentences before each of these two paragraphs can frame the context to make it more obvious why the reader should know these details. Along the same lines, the paragraph about cytokines is confusing, because there are dozens of cytokines and many many functions. It's not clear why this particular aspect of cytokines is being highlighted, and in particular why this is being highlighted as the concluding paragraph to the section. Finally, There are a lot of details leading up to line.

We have rewritten this paragraph like this: " In addition to the canonical caspase 1-dependent pathway, the inflammasome can be activated through caspase 4- and 5-dependent pathways, known as the non-canonical pathway. This noncanonical caspase-4/5 inflammasome is activated by intracellular LPS and requires NLRP3 for its activation. It plays a crucial role in supervising cytosolic gram-negative bacteria by sensing lipopolysaccharide (LPS). Extracellular LPS generated by gram-negative bacteria is internalized via TLR4- or RAGE/HMGB1-mediated endocytosis. Subsequently, guanylate-binding proteins (GBPs), which are expressed in response to interferons and other proinflammatory cytokines, alter the integrity of the endosomal membrane, thereby promoting the release of LPS into the cytoplasm. Cytosolic LPS binds directly to caspase 4-5, which then dimerizes to the active form and leads to the cleavage of GSDMDs, activation of the canonical NLRP3 inflammasome [43]. “

  1. Can you add a concluding paragraph that ties things together into one framework? This would make for a more natural transition to the next topic, DNA damage and repair in inflammation.

We have added: “Generally, organisms possess intricate systems that identify and eradicate pathogens, protect against external infections, and maintain tissue homeostasis. The activation of these systems triggers an acute inflammatory response that involves the remodeling of local tissue through the release of cytokines that attract immune cells and regulate their functions (48). Acute inflammation is beneficial because it can quickly resolve tissue damage caused by pathogens or internal stimuli. However, if acute inflammation fails to eliminate the causes of inflammation and repair tissue damage, the inflammatory response persists, becomes chronic, and develops new characteristics that can damage DNA and compromise tissue health [48]. Chronic inflammation does not appear to be caused solely by infections or injuries, but rather by malfunctions in tissue due to factors such as, DNA repair deficiency, nutrition, epigenetic remodeling, and cancer.”

P4 line 179 should read “into the cytoplasm

We have corrected it

 P4 line 180: Also caused by nondisjunction?

 We have added non disjunction

P5 line 181: there doesn’t need to be a DNA repair defect; there can just be exposure to a DNA damaging agent. Also, there is a lot known about how DNA released from micronuclei stimulates cGAS. This should be emphasized as the more important of the processes by which Mn invoke the IFN response.

We have modified the text like this: “These defects can be caused by mutations in genes involved in the DNA damage response (DDR) and the regulation of the mitotic spindle, or they can be induced by DNA-damaging treatments [52-54]”

We have added this sentence: “Although the c-GAS-STING pathway serves as the primary mechanism to induce inflammatory responses to genotoxic stress, DDR proteins can also trigger inflammation by directly or indirectly activating NF-kB”

 P5 line 197: this paragraph is misleading because NFkB can be activated independently from ATM and ATR. This can happen though the cGAS-STING pathway.

We have specified  that:” The migration of chromosomal DNA fragments from micronuclei to the cytoplasm results in the activation of cGAS, which subsequently leads to the synthesis of cGAMP. This, in turn, triggers the inflammatory process by inducing the production of type I interferon and cytokines via the STING-TBK1-IRF3 and STING-TBK1-NFkB signaling pathways, respectively”

P5 Line 194: Reference 57 shows that STING is activated in an ATM-dependent fashion, but does not show a direct interaction between ATM and STING. This reviewer recommends re-reading references 57 and 58 to ensure accuracy of this paragraph.

We have corrected the sentence like this: “ Damaged DNA, induced by genotoxic agents, recruits and activates ATM and ATM and RAD3-related (ATR) which can stimulate NF-kB activation through: 1) stabilization of GATA4 through inhibition of p62 and autophagic degradation of GATA4 [57] ; 2) assembly of an alternative STING signaling complex including the tumor suppressor p53 and the E3 ubiquitin ligase TRAF6[58]; 3) degradation of IκBα through the formation of the IκBα-β-TrCP-ubiquitin ligase complex and phosphorylation of RELA (also known as p65) through interaction with protein kinase A (PKA)[59].”

P5 Line 197: This paragraph is very difficult to read. It is a long list of facts, but little framing and many different topics in one very long paragraph. First, a topic sentence is needed. Second, much of the content of this paragraph needs to be separated out to make it more readable. There are at least three separate threads in this paragraph: ATM, NER, NHEJ. It is suggested that these be parsed out. Also, Line 229 has a typo: should read “NHEJ”.

We have rewritten and restructured this paragraph like this:

“3.1. Mechanism underlying DNA damage induced inflammation. DNA repair mechanisms and signaling pathways are closely linked to the inflammatory response. DNA damage activates the cGAS-STING pathway, which………

3.2.DNA repair deficiency disorder and inflammation. The DDR is a vital mechanism that ensures the preservation of genomic integrity and prevents tumor formation by employing an intricate signaling network that detects, signals, and repairs DNA damage…….

3.3. Inflammation promoted by Transposable Elements (TEs) instability. Dysfunctions in the DNA Damage Response pathway can result in a strong inflammatory…….

3.4. Inflammation triggered by RLoops resolving defects. Inflammation can arise from disruptions in the mechanisms responsible for maintaining and…….

3.3. Anti-tumor cytotoxicity mediated by inflammatory response. The effectiveness of chemotherapeutic agents, including anthracyclines….”

P5 line 197: Conspicuously absent is a description of the fact that defects in polymerase beta leads to an interferon response (PMID: 36624848).

We have added: “Even the inhibition of proteins involved in base excision repair (BER) can elicit an interferon response through activation of the cGAS pathway. Mutations in the DNA polymerase beta (POLB) protein, which is involved in BER and fills in single-nucleotide gaps, result in replication-associated genomic instability and inflammation-associated carcinogenesis in mice. This occurs because failure to repair the damage leads to the formation of single-strand breaks (SSBs), which are converted into double-strand breaks (DSBs) during the S-phase of DNA replication. The accumulation of DSBs causes mitotic dysfunction-, leading to an increase in micronuclei formation due to the mis-segregation of broken chromosomes during mitosis. This, in turn, triggers a c-GAS-mediated inflammatory response by releasing cytosolic DNA[72].”

Figure 3 is confusing. The bubble on the top right is nice, and explains that DNA gets methylated. But, the bubble on the top left seems to suggest that HDAC/HAC is a process that is independent of 5MeC, which is not necessarily the case. Further, it’s not clear why top left bubble is indicated as “metabolic reprogramming” while the 5MeC story is labeled as epigenetic modification.

In figure 3, we present the epigenetic mechanisms without connecting them sequentially. Moreover, the arrows in the circle indicate that metabolism, inflammation, and epigenetics are interrelated, and the external arrows generally signify the general epigenetic mechanisms that inflammation and metabolism can influence.

P7 Figure 3 caption: There is no mention of miRNA in the caption.

We have added miRNA description in the caption

Page 5: The relationship between 5meC and HDAC is missing.

We have added:” Transcriptional regulation is tightly controlled by coordinated action of histone and DNA modification proteins. Methylated DNA interacts with the methyl-CpG-binding protein (MBD) family, which facilitates histone deacetylation by attracting various transcriptional corepressor complexes that contain HDACs[120]. DNMT1 and DNMT3a limit gene expression by binding to SUV39H1, an enzyme that methylates H3K9 [121]. Additionally, DNMT1 and DNMT3b interact with HDACs and regulate gene expression [122, 123]. While the precise mechanism governing the interaction between histone acetylation and DNA demethylation remains elusive, it is worth noting that two studies have revealed that suppressing HDAC triggers active DNA demethylation [124, 125]”

Page 5: there is a lot of text describing processes related to epigenetic regulation, which is probably too detailed to fit into this particular review. By way of contrast, very little text is devoted to explaining DNA repair pathways and instead the text is focused on how DNA repair-related proteins affect inflammation. This is fine, but this is in sharp contrast to the section on epigenetics.

That said, if this is kept as is, there needs to be a subheaders since the flow goes from describing epigenetic processes to describing how epigenetic processes impact inflammation without any transition statement.

I have chosen not to incorporate a description of the DNA repair mechanism in this paragraph. However, I have included the specified sbheaders to provide a comprehensive overview of the topic.

4.1. Epigenetic regulatory mechanisms

Histone modifications……

DNA methylation….

Non coding RNA….

4.2. Epigenetic signatures underlying inflammation

Histone acetylation….

Histone Methylation…..

DNA methylation….

Non coding RNA….

4.3. Epigenetic modulation of the immune function.”

P9 Line 364 states that NFkB dependent gene expression depends on histone acetylation. This is confusing because it makes it seem like NFkB’s activity is controlled by histone acetylation, when it is the other way around. Histone acetylation opens the chromatin and makes it accessible to NFkB. This should be made clear

We have added  this sentence:” NF-κB-dependent gene expression requires the involvement of histone acetylase which will decompress the repressed chromatin environment through histone acetylation[130, 131].”

Page 9: There are a lot of details here, but what is missing is a bigger picture perspective.

We have added this final sentence which summarizes the concepts expressed in the paragraph:

“Epigenetic mechanisms are crucial in regulating inflammatory responses and in the functioning and differentiation of immune cells. In pathological situations, epigenetic reprogramming can worsen physiological conditions, promoting the advancement of diseases. Nevertheless, these modifications are reversible, making them attractive therapeutic targets in a variety of diseases. The regulation of these mechanisms is impacted by both internal and external factors, one of which is diet.”

Page 9 line 397: This is confusing. LPS triggers TLRs that ultimately activate NFkB. This sentence suggests that p65 (part of NFkB) is expressed when LPS causes demethylation. This seems unlikely because NFkB bound to IkkB is already in the cytosol, and it is release of NFkB from its inhibitor that leads to it becoming an active transcription factor. Certainly silencing of NFkB reduces its ability to act as a transcription factor. If this is the intended message, this should be made clear.

We have rewritten the sentence like this: ”KDM7C regulates proinflammatory cytokines like TNF, CCL4, and Il11 upon TLR-4 ac-tivation by LPS. TLR-4 stimulates NF-kB, which then recruits and delivers JHDM1E to the promoter region of these genes, where it demethylates H4K30me3 to initiate transcription [142]”

P10 Line 399: This paragraph needs a topic sentence. Also, cause and effect is confusing in the existing first sentence.

We rewrite the sentence like this: “Histone methyltransferases also suppress PRR signal transduction. Following papilloma virus infection, the E7 oncoprotein forms a complex with NF-κBp50–p65 and recruits histone demethylase JARID1B and histone deacetylase HDAC1. This complex binds to a specific region on the TLR9 promoter, leading to decreased methylation and acetylation of histones upstream of the TLR9 transcriptional start site[144]. The overexpression of KDM5B/5C histone demethylases has been found to have a detrimental impact on the innate immune response in breast cancer.

P14 Line 599. This section is more readable than prior parts of the review, but the flow of ideas seems disjointed. It seems that many paragraphs are a stand-alone paragraphs and it not clear why the paragraphs are organized as they are. Perhaps subheaders would help.

This paragraph has been completely restructured by adding subheaders and rewritten in some parts, as can be seen directly in the text.

P15 669: This is a two-sentence paragraph that reads more like a list of facts rather than a paragraph that is framed, substantiated, and that has a conclusion. The second half of page 20 also has a series of very short paragraphs that would benefit from framing.

This paragraph has been completely restructured by adding subheaders and rewritten in some parts, as can be seen directly in the text.

We have made all the suggested changes to Figure 1

Reviewer 2 Report

Comments and Suggestions for Authors

Manuscript certainly deserves publication as it appropriately reviews most major factors involved in the etiopathogenesis of malignant neoplastic diseases. However, there are some improvements that should be done before it is published. Text content, and especially conclusion, should more completely prove what it is stated in Title and that is the major message that could be given. It is claimed that a “web” of interconnections links inflammation, epigenetics, and nutrition in cancer. Beside the fact that using the word “killer” is quite rude talking about disease conditions affecting people with pain and distress, although this might occur only for an individual sensitivity, the major point is that manuscript mostly and accurately describes mechanisms involved in the development of neoplastic diseases, with an actual limited description of their interconnections. In addition to the analytical report of the neoplastic-related mechanisms of inflammation, epigenetics and nutrition, deeper and wider discussion should be provided to consider on how these factors could network and cooperate negatively or positively, especially when therapeutic perspectives are then presented.

Author Response

Dear Reviewer,

I would like to express my heartfelt gratitude for your insightful suggestions that have greatly improved the quality and clarity of our paper. To enhance the readability and comprehension of the manuscript, we have carefully organized the review into subchapters, each ending with a concluding thought. In addition, we have thoroughly revised and reorganized the section on the role of diet in oncogenesis, as well as its interplay with inflammation and epigenetics. We have also expanded upon the section regarding the use of dietary interventions in the prevention and treatment of tumors. We hope that these changes have addressed your concerns and have made the review suitable for publication. Regarding the title, we understand that it may be perceived as sensitive by some readers, but our intention is only to evoke the title of a popular film. To facilitate identification of the modified sections, we have highlighted them in red.

Thank you for taking the time to review our manuscript.

Best regards

Round 2

Reviewer 1 Report

Comments and Suggestions for Authors

Overall, this comprehensive review is an important contribution to the field. It is well written and detailed with appropriate references. This reviewer is enthusiastic about this manuscript.

Of note, inflammation also leads to high levels of reactive oxygen and nitrogen species, including superoxide, hydroxyl radical and peroxynitrite. The RONS can damage the DNA, promoting inflammation in a feed forward mechanism. The fact that inflammation is associated with induction of DNA damage is a big missing piece. This is especially the case for chronic inflammation, which can be pro-mutagenic, thus potentiating cancer.

There are numerous errors in Figure 1. For example, MAPK is upstream of NFkB, not at the same level as is what is in the figure. IKK-beta is not upstream of MAPK. IKK-beta is upstream of NFkB, which is not shown. The authors need to reconfirm all of the connections in Figure 1 to ensure accuracy. Many readers will rely heavily on this figure specifically.

HMGB1 is released by either passive release (necrosis) or active release.

Page 5 line 204: cGAS STING is also stimulated by extracellular DNA.

Page 8 line 346: PARPi do not induce DNA damage. They inhibit DNA repair which leads to DNA damage. Also, they are not specific to ERCC1 deficient cells. PARPi induce their effects in any cell type, and are particularly toxic for cells deficient in homology directed repair, such as BRCA2 null cells.

Line 351: It would be good to also mention that the IFN response is critical to the efficacy of radiation in the treatment of cancer, either here or on page 27. See papers by Au.

Line 412: What is missing from this paragraph is the link between HDACs and 5MeC. Either here or below, there should be mention that HDACs are associated with proteins that bind to 5MeC, allowing for selective suppression of gene expression.

Line 679: What is meant by ‘living beings’ (which ones? Name a few) and what specifically is known for humans about the impact of CR?

1042: p53 has many functions independent of its effects on DNMTs. It is the most frequently mutated gene probably because without it, cancer cells become resistant to DNA damage-induced apoptosis.

Line 1195: This section belongs with the section on caloric restriction, since it really is separate from issues of nutrition. How does one differentiate “overnutrition” from excess calories and resulting increase in fat that then affects inflammation? Likewise, for CR, is it the reduction in metabolism or is it the reduction in nutrients?

1211: what is an “unbalanced diet”?

1266: It seems that this aspect makes sense specifically in the case of chronic inflammation.

1305: This is more general than just PARPi. This paragraph applies to many types of DNA damage such as that that is induced by conventional chemotherapy or radiation. It would also be good to remind the reader of what was written earlier, which is that defects in DNA repair can lead to increased levels of DNA damage which in turn can promote inflammation.

Author Response

Below are point-by-point responses to the reviewer's comments.

Overall, this comprehensive review is an important contribution to the field. It is well written and detailed with appropriate references. This reviewer is enthusiastic about this manuscript.

  • Of note, inflammation also leads to high levels of reactive oxygen and nitrogen species, including superoxide, hydroxyl radical and peroxynitrite. The RONS can damage the DNA, promoting inflammation in a feed forward mechanism. The fact that inflammation is associated with induction of DNA damage is a big missing piece. This is especially the case for chronic inflammation, which can be pro-mutagenic, thus potentiating cancer.

We added this sentence:

“The production of reactive oxygen species (ROS) and oxidative nitrogen species, such as superoxide, hydroxyl radical, and peroxynitrite, by myeloid cells during inflammation in the tumor microenvironment causes DNA damage. This, in turn, activates the previously mentioned pro-inflammatory pathways, further fueling inflammation. This crosstalk between inflammation and DNA damage creates a feedback loop that promotes tumor progression”.

  • There are numerous errors in Figure 1. For example, MAPK is upstream of NFkB, not at the same level as is what is in the figure. IKK-beta is not upstream of MAPK. IKK-beta is upstream of NFkB, which is not shown. The authors need to reconfirm all of the connections in Figure 1 to ensure accuracy. Many readers will rely heavily on this figure specifically.

We have made adjustments to the figure, as requested.

  • HMGB1 is released by either passive release (necrosis) or active release.

We have modified this sentence and specified that HMGB1 can be released passively or actively.

“Inflammasome activation triggers the release of proinflammatory cytokines and damage-associated molecular patterns (DAMPs), including high-mobility group box 1 (HMGB1). HMGB1 can be released into the extracellular space by both actively secreted by inflammatory and immune cells and passively secreted by damaged or necrotic cells with compromised plasma membranes.”

  • Page 5 line 204: cGAS STING is also stimulated by extracellular DNA.

We specified that cGAS is activated by endogenous and exogenous DNA

“cGAS triggers this process by detecting endogenous DNA released from the nucleus, mitochondria, or micronuclei into the cytoplasm or as well as exogenous DNA derived from pathogenic microorganisms.”

  • Page 8 line 346: PARPi do not induce DNA damage. They inhibit DNA repair which leads to DNA damage. Also, they are not specific to ERCC1 deficient cells. PARPi induce their effects in any cell type, and are particularly toxic for cells deficient in homology directed repair, such as BRCA2 null cells.

We eliminated "PARPi-induced inflammatory responses":

“As a result, interferon-stimulated genes (ISGs) are expressed in BRCA1/2-deficient tumor cells, and stimulate T cell infiltration and activation, ultimately leading to tumor eradication in BRCA1/2-deficient mouse models of ovarian and breast cancer”

  • Line 351: It would be good to also mention that the IFN response is critical to the efficacy of radiation in the treatment of cancer, either here or on page 27. See papers by Au.

We added this sentences and Auh reference.

“Notably, the release of damaged DNA fragments and the accumulation of micronuclei in the cytosol, which are induced by radiotherapy, stimulate the production of IFN-I via the cGAS pathway. This, in turn, promotes the maturation of dendritic cells (DC), enhancing their cross-priming capacity and ultimately leading to the activation of T cells.”

  • Line 412: What is missing from this paragraph is the link between HDACs and 5MeC. Either here or below, there should be mention that HDACs are associated with proteins that bind to 5MeC, allowing for selective suppression of gene expression.

This concept was discussed in the line:

Line 466: Transcriptional regulation is tightly controlled by coordinated action of histone and DNA modification proteins. Methylated DNA interacts with the methyl-CpG-binding protein (MBD) family, which facilitates histone deacetylation by attracting various transcriptional corepressor complexes that contain HDACs[120]. DNMT1 and DNMT3a limit gene expression by binding to SUV39H1, an enzyme that methylates H3K9 [121]. Additionally, DNMT1 and DNMT3b interact with HDACs and regulate gene expression [122, 123]

  • Line 679: What is meant by ‘living beings’ (which ones? Name a few) and what specifically is known for humans about the impact of CR?

We changed the sentence like this:

“various organisms, ranging from invertebrates such as worms and flies to vertebrates including rodents and primates.”

  • 1042: p53 has many functions independent of its effects on DNMTs. It is the most frequently mutated gene probably because without it, cancer cells become resistant to DNA damage-induced apoptosis.

We added:

“p53 is the most commonly mutated tumor suppressor gene, and its alterations or loss can significantly impact epigenetic reprogramming as well as rendering tumor cells resistant to apoptosis.”

  • Line 1195: This section belongs with the section on caloric restriction, since it really is separate from issues of nutrition. How does one differentiate “overnutrition” from excess calories and resulting increase in fat that then affects inflammation? Likewise, for CR, is it the reduction in metabolism or is it the reduction in nutrients?

We added this sentence:

“Ovenutrition refers to the intake of excessive amounts of both macro- and micronutrients, which are then stored in the body's tissues, especially adipose tissue. When there is no longer any space to store these excess nutrients, adipocytes swell and undergo changes that result in chronic inflammation which can lead to noncommunicable diseases such as diabetes mellitus, coronary artery disease and stroke. Overnutrition is considered a form of malnutrition due to its detrimental impact on health. It is distinct from a high-calorie diet, which is often linked to fast food and is rich in macronutrients but poor in micronutrients, resulting in a low nutritional value.”

Regarding the second question, calorie restriction simply involves decreasing calorie intake without altering the micronutrient content. When calorie intake is restricted for an extended period, it results in a decrease in basal metabolism as a consequence of epigenetic reprogramming.

  • 1211: what is an “unbalanced diet”

We specified:

“An unbalanced diet (excess or deficiency of macro or micronutrients)”

  • 1266: It seems that this aspect makes sense specifically in the case of chronic inflammation.

We added:

“Non-steroidal anti-inflammatory drugs (NSAIDs) may offer a valuable strategy for combating inflammatory cancer by targeting chronic inflammation.”

  • 1305: This is more general than just PARPi. This paragraph applies to many types of DNA damage such as that that is induced by conventional chemotherapy or radiation. It would also be good to remind the reader of what was written earlier, which is that defects in DNA repair can lead to increased levels of DNA damage which in turn can promote inflammation.

We added:

“Preclinical findings suggest that the use of suboptimal doses of DNA-damaging agents in combination with ICIs or other immunotherapeutics can effectively stimulate tumor-targeting immune responses by increasing the inflammatory response via cGAS activation.”

We would like to express our gratitude to the reviewer for providing us with valuable feedback, and we are confident that the modifications we have made will ensure that the article is suitable for publication.